# Comparative evaluation of Olink Explore 3072 and mass spectrometry with peptide fractionation for plasma proteomics
Noora Sissala [1] ✉, Haris Babačić [1], Isabelle R. Leo [1,2], Xiaofang Cao[1], Jenny Forshed[1,3], Lars E. Eriksson [4,5], Janne Lehtiö [1], Claudia Fredolini [6,7], Mikael Åberg [8,9] & Maria Pernemalm [1] ✉

Plasma proteomics technologies are advancing rapidly, offering new opportunities for biomarker discovery and precision medicine. Direct comparisons of available technologies are needed to understand how platform selection affects downstream findings. We compared the performance of a peptide fractionation-based mass spectrometry method (HiRIEF LC-MS/MS) and the Olink Explore 3072 proximity extension assays on 88 plasma samples, analyzing 1129 proteins with both methods. The platforms exhibited complementary proteome coverage, high precision, and concordance in estimating sex differences in protein levels. Quantitative agreement between platforms was moderate (median correlation 0.59, interquartile range 0.33-0.75), mainly influenced by technical factors. Finally, we present a publicly available tool for peptide-level analysis of platform agreement and demonstrate its utility in clarifying cross-platform discrepancies in protein and proteoform measurements. Our findings provide insights for platform selection and study design, and highlight the value of combining mass spectrometry and affinity-based approaches for more comprehensive and reliable plasma proteome profiling.

Proteins are the primary effector molecules of cells and tissues, and their levels closely reflect the phenotype and physiological state of an individual. The plasma proteome, comprising a complex mixture of proteins from virtually all organs in the body, represents a rich source of biological information and is readily accessible through a simple blood test. These features have motivated efforts to profile the plasma proteome in diverse conditions, with applications ranging from elucidating disease mechanisms to biomarker discovery and precision medicine[1,2].

Yet, the complexity of the plasma proteome makes it challenging to analyze. Protein concentrations in plasma span at least 10 orders of magnitude, and among the thousands of proteins present, the 22 most abundant constitute 99% of the total protein mass[3]. Disease-related proteins are often present at low levels, and their detection has historically required either extensive sample processing for untargeted analysis or targeted analysis of individual or small sets of proteins. However, recent advancements in both global mass spectrometry (MS) and highly multiplexed affinity-based proteomics technologies have alleviated this problem by simultaneously increasing proteome coverage and sample throughput[4,5]. Consequently, larger cohorts can be profiled comprehensively, increasing the potential for insights into human health and disease and facilitating the discovery of new biomarkers.

In global MS-based approaches, proteins are measured in an untargeted manner by digesting proteins into peptides, separating and ionizing the peptides, measuring their mass-to-charge ratios with MS, and identifying and quantifying the peptides by matching their mass spectra to theoretical mass spectra from sequence databases (peptide-spectrum matching). Generally, MS-based approaches offer highly specific identification and quantification of detected proteins (peptides) but usually require several steps in sample preparation for in-depth profiling. Some MS methods include fractionation, i.e., separation of peptides into fractions based on physicochemical properties, to achieve greater depth, often at the cost of analysis time[6,7]. Thus, studies employing in-depth MS-based proteomics have been limited in their sample size compared to those employing affinity-based methods[7-10].

[1]Department of Oncology-Pathology, Karolinska Institutet and Science for Life Laboratory, Solna, Sweden. [2]Department of Immunobiology, Yale School of Medicine, New Haven, CT, USA. [3]Capio Elderly and Mobile Care, Stockholm, Sweden. [4]Department of Neurobiology, Care Sciences and Society, Karolinska Institutet, Huddinge, Sweden. [5]School of Health and Medical Sciences, City St George's, University of London, London, United Kingdom. [6]Department of Protein Science, School of Engineering Sciences in chemistry, Biotechnology and Health, KTH Royal Institute of Technology, Stockholm, Sweden. [7]Affinity Proteomics Unit, Science for Life Laboratory, Solna, Sweden. [8]Department of Medical Sciences, Uppsala University, Uppsala, Sweden. [9]Affinity Proteomics Unit, Science for Life Laboratory, Uppsala, Sweden. ✉e-mail: noora.sissala@ki.se; maria.pernemalm@ki.se

In contrast, affinity-based approaches use affinity molecules such as antibodies or aptamers to bind and quantify pre-defined target proteins, enabling high-throughput profiling of the plasma proteome[2]. Olink's antibody-based proximity extension assays (PEAs) and SomaLogic's aptamer-based SomaScan assays have facilitated large-scale studies involving thousands of individuals[11,12]. However, unlike MS, these methods do not provide direct detection of proteins (peptides), and ensuring the specificity and accuracy of affinity binders is challenging. To help mitigate this issue, PEAs rely on two antibodies to detect each target protein[13].

Given the differing strengths and limitations of plasma proteomic platforms, understanding how platform selection influences protein quantification, reproducibility, and biological interpretation is critical for guiding study design. While the performance of Olink's PEAs and SomaLogic's SomaScan assays has been compared extensively[14–26], direct comparisons with MS remain scarce and have generally been limited in profiling depth[25–28]. Here, we present a comprehensive comparative evaluation of the Olink Explore 3072 PEA-based platform and our previously published method for in-depth MS-based plasma proteomics, which combines high-resolution isoelectric focusing with liquid chromatography-tandem MS (HiRIEF LC-MS/MS)[7]. This workflow involves depletion of high-abundance proteins, tandem mass tag (TMT) labeling, extensive pre-fractionation of peptides using HiRIEF, and data-dependent acquisition (DDA) to achieve high analytical depth and relative quantification. We evaluate the two methods in terms of proteome coverage, precision, statistical power, and quantitative agreement at both the protein and peptide level. Finally, we present PeptAffinity, a publicly available tool for visualizing peptide-level agreement between HiRIEF LC-MS/MS and Olink Explore 3072 along the protein sequence and structure. We demonstrate the utility of PeptAffinity in enabling a more detailed investigation of cross-platform discrepancies in protein quantification, revealing differential proteoform measurement.

## Results
### Study overview

We detected 2578 unique proteins across 120 samples using HiRIEF LC-MS/MS (114 distinct samples with six samples run in duplicate) and measured 2923 proteins (2941 distinct Olink assays) in a subset of 88 samples using Olink Explore 3072 (Fig. 1). For Olink, normalized protein expression (NPX) values below the limit of detection (LOD) were retained in all analyses, unless stated otherwise. Ten proteins with NPX values below the LOD in all samples were deemed not detected and were excluded from further analysis. In total, 4362 proteins were detected and quantified in at least one sample across both technologies, 2578 with MS and 2913 with Olink, with 1129 overlapping between methods (Fig. 2A, Supplementary Data 1). The number of overlapping proteins varied by Olink Explore panel, with the greatest overlap observed for the Cardiometabolic panel (Fig. 2B). The frequency of missing values (missing in MS or <LOD in Olink) differed between platforms. In the MS data, 53% of all quantified proteins had at least one missing value, compared to 35% of proteins in the Olink data (Fig. 2C). A total of 1741 proteins were detected in at least 50% of the 88 samples analyzed with both technologies using MS, and 2460 using Olink, while 1212 and 1910 were detected in all samples using MS and Olink, respectively. In the MS data, missing values were TMT set-specific (Fig. S1).

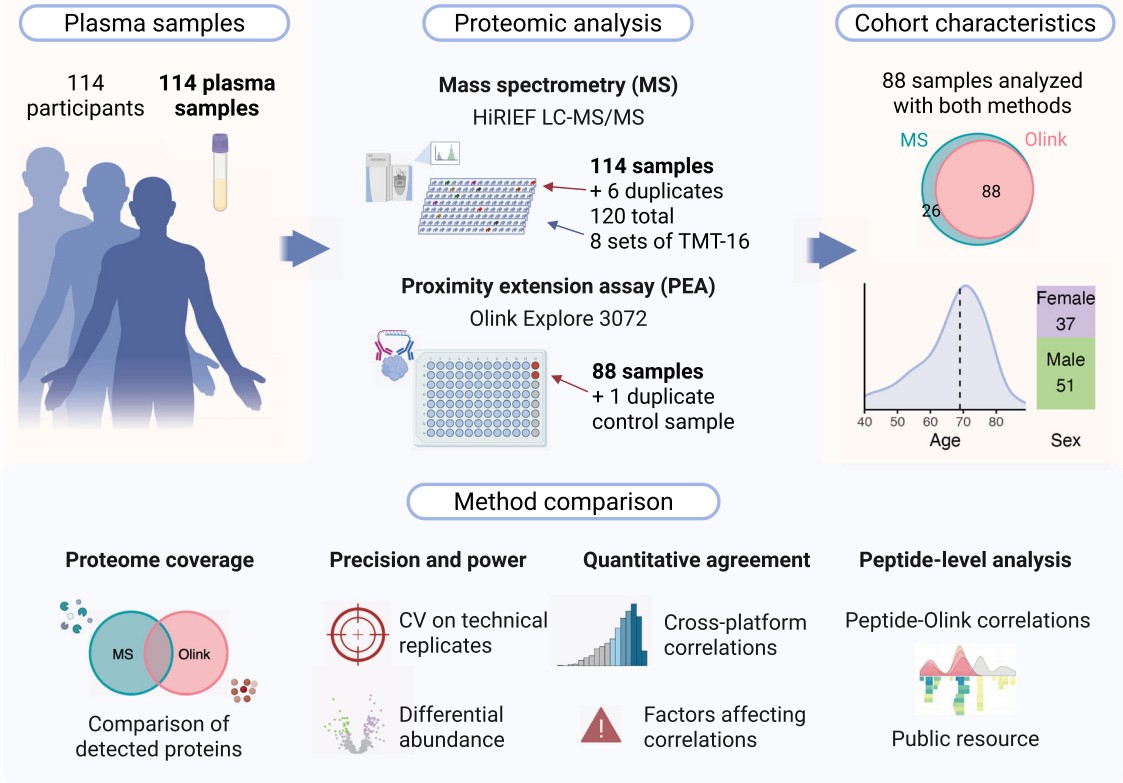

**Fig. 1 | Overview of the study.** Plasma samples: a total of 114 pre-diagnostic plasma samples were collected from patients under investigation for suspected lung cancer. Proteomic analysis: all 114 samples were analyzed using mass spectrometry (MS)-based proteomics (HiRIEF LC-MS/MS). Six samples were run in duplicate in different tandem mass tag (TMT) sets, resulting in a total of 120 samples for the MS analysis. A subset of 88 samples was also analyzed using the Olink Explore 3072 proximity extension assays (PEAs), along with one duplicate control sample. Duplicate samples were used to estimate analytical precision by calculating technical coefficients of variation (CVs). Cohort characteristics: age and sex distribution of the study population (N = 88). Method comparison: The methods were compared in terms of proteome coverage, precision, statistical power and concordance in detecting differential protein abundance. Quantitative agreement between HiRIEF LC-MS/MS and Olink Explore 3072 measurements was assessed at both the protein and peptide levels. A publicly available resource, the PeptAffinity R Shiny app, was developed for exploring peptide-level agreement along the protein sequence and structure. Created in BioRender (https://BioRender.com/i93h320).

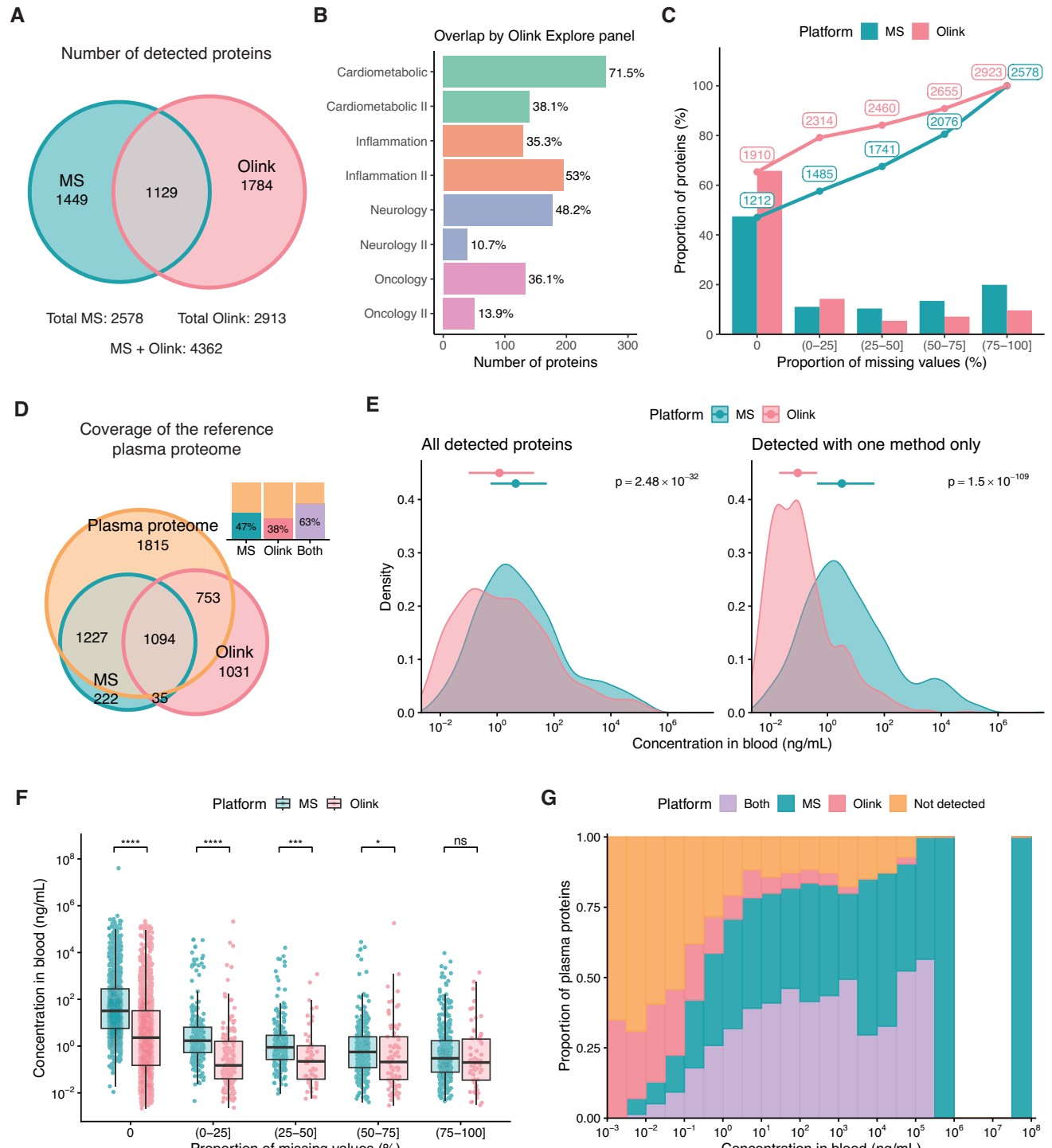

**Fig. 2 | Detected proteins, missing values, and plasma proteome coverage. A** Venn diagram of proteins detected in at least one sample with HiRIEF LC-MS/MS and/or Olink Explore 3072, based on unique UniProt IDs. **B** Number and percentage of Olink assays in each Olink Explore panel detected with both Olink and MS. **C** Detected proteins and missing values in the MS and Olink datasets. The y axis indicates the percentage of proteins in each dataset with a percentage of missing values within the intervals defined on the x axis. The dotted line shows the cumulative number of proteins with a proportion of missing values less than or equal to the upper bound of each interval. Percentages were calculated out of 88 samples analyzed with both methods. **D** Venn diagram comparing proteins detected by MS and Olink to proteins in the reference human plasma proteome, compiled from the Human Plasma Proteome Project (HPPP) database and the Human Protein Atlas (HPA). The bar plot shows the proportion of proteins in the reference plasma proteome detected with MS, Olink, or both methods. **E** Distribution of estimated concentrations, from the HPA, of all detected proteins (left) and proteins detected exclusively by MS or Olink (right). Medians and interquartile ranges are indicated with points and error bars. Differences in protein concentration between platforms were tested using a two-sided Wilcoxon rank-sum test. **F** Missing values per protein by estimated protein concentration. Differences in protein concentration between platforms were tested using a two-sided Wilcoxon rank-sum test, and p values were adjusted using the false discovery rate method. ns = not significant, *$p < 0.05$, ***$p < 0.001$, ****$p < 0.0001$. **G** Plasma proteome coverage by estimated protein concentration. Each bar shows the proportion of proteins in the reference plasma proteome, within a specific concentration interval, that were detected with either MS only, Olink only, both methods, or neither method ("Not detected"). The x axis intervals are right closed.

## Plasma proteome coverage

To assess the plasma proteome coverage of each method, we first curated a reference set of 4889 plasma proteins by compiling proteins from the Human Plasma Proteome Project (HPPP, www.peptideatlas.org)[29] and the Human Protein Atlas (HPA, www.proteinatlas.org) (see Methods)[30,31]. HiRIEF LC-MS/MS showed a greater overlap with this reference plasma proteome, while Olink Explore 3072 measured more than a thousand proteins not reported in the MS-based studies found in the HPPP (Fig. 2D, Supplementary Data 1). Combined, the platforms covered 63% of the reference plasma proteome.

Based on blood concentration estimates from the HPA[31], both technologies detected proteins with concentrations spanning 10 orders of magnitude, down to picograms per milliliter (Fig. 2E). However, low-abundance proteins frequently had a large proportion of missing values, especially in the MS data (Fig. 2F). Olink demonstrated a higher coverage of low-abundance proteins, while MS demonstrated a higher coverage of mid to high-abundance proteins (Fig. 2G). Therefore, proteins detected exclusively by Olink were mainly low abundance, whereas those detected exclusively by MS tended to be of higher abundance (Fig. 2E). These observations remained consistent when considering proteins detected in at least 50% of samples, although the coverage of low-abundance proteins decreased for both methods (Fig. S2).

## Characterization of detected proteins

Based on HPA annotations, predicted secreted proteins, enzymes, metabolic proteins, immunoglobulins, proteins enriched in liver tissue, and potential drug targets were more frequent in the MS data, while predicted membrane proteins, CD markers, proteins secreted in the male reproductive system, and proteins enriched in the brain and testis were more frequent in the Olink data (Fig. 3A, Supplementary Data 2). Notably, 95 proteins (3.7%) detected by MS were not found in the HPA, compared to only 22 proteins (0.76%) detected by Olink, which could lead to a slight underestimation of annotation frequencies for MS. As expected, MS was enriched for Gene Ontology (GO) biological processes related to high-abundance plasma proteins—hemostasis, blood coagulation, complement activation, and metabolism, while Olink was enriched for processes related to low-abundance signaling proteins, particularly cytokines (Fig. 3B, Supplementary Data 3). The methods detected comparable numbers of United States food and drug administration (FDA)-approved plasma protein biomarkers[32]—74 (MS) and 72 (Olink) out of 99, with 55 biomarkers detected by both (Supplementary Data 4). Biomarkers exclusively detected by MS included various transport and metabolic proteins, whereas Olink exclusively covered various hormones.

## Precision

To evaluate the precision of repeated measurements with HiRIEF LC-MS/MS and Olink Explore 3072, we calculated technical coefficients of variation (CV) for each protein across duplicate samples (Supplementary Data 5). For Olink, intra-assay CVs were derived from a control sample of pooled donor plasma run in duplicate on the same plate. Data for one control were missing for a subset of assays due to a technical failure, leaving 2197 protein assays (2185 unique proteins, 75%) for CV calculation. For MS, inter-assay CVs were calculated using duplicates of patient samples, with each replicate run in different TMT sets. Due to variable protein identifications between TMT sets (i.e., missing values), CVs could be calculated for 1952 proteins (76%).

The platforms demonstrated high precision (Fig. 4A), with comparably low technical CVs for both MS (median: 6.8%, mean: 9.4%) and Olink (median: 6.3%, mean: 9.8%). Most proteins had CVs below 15% in both datasets (MS: 85%, Olink: 81%), although Olink had more proteins with very low CVs, below 5% (MS: 33%, Olink: 41%). However, the Olink CVs might have been underestimated, since these were intra-assay CVs, while for MS, we calculated inter-assay CVs. Technical CVs were higher for proteins with more missing values and lower estimated blood concentrations (Fig. S3).

For MS, we also calculated technical CVs for 27,462 peptides mapping to the 2578 detected proteins. As expected, technical CVs were somewhat higher at the peptide level, with a median of 10.2% (Fig. 4B).

## Statistical power

To explore differences in statistical power and their impact on the consistency of biological insights provided by HiRIEF LC-MS/MS and Olink Explore 3072, we performed differential abundance analyses (DAA) between males and females and compared the differentially abundant proteins (DAPs) identified. When considering all overlapping proteins in the analysis ($N = 1129$), we identified 76 DAPs in the MS data and 180 DAPs in the Olink data, with 50 (24%) identified in both (Fig. 4C, Supplementary Data 6). While the overlap in statistically significant DAPs was modest, the platforms showed strong concordance in estimated differences between the groups (Fig. 4D). The lack of overlap in statistical significance could be partly explained by differences in data completeness—more than half of the DAPs identified exclusively by Olink had missing values in the MS data, resulting in a smaller sample size and lower statistical power for these proteins in the MS analysis (Fig. S4A). To ensure equal sample sizes, we next restricted the analysis to overlapping proteins with no missing values or values < LOD ($N = 569$). In this setting, we identified 82 and 118 DAPs in the MS and Olink data, respectively, with 53 (36%) identified in both (Fig. 4E, Supplementary Data 7). Although fewer proteins were analyzed, the number of DAPs in the MS data increased due to a less stringent correction for multiple hypothesis testing. Notably, the agreement in estimated differences also improved, with a correlation of $R = 0.93$ and 95% directional agreement for proteins significant in at least one of the platforms (Fig. 4F). These results demonstrate that despite the differences in the number of DAPs identified, both platforms captured consistent biological signals.

We further investigated the effect of technical and biological variance on statistical significance. DAPs identified only by Olink tended to have higher technical CVs in the MS data, but not vice versa (Fig. S4B), suggesting that technical noise in MS may have masked some protein-level differences between the sexes. Moreover, the log2-fold change estimates for DAPs were generally larger in the Olink data (Fig. S4C), and DAPs unique to Olink had lower estimated concentrations in the blood than those shared with or unique to MS (Fig. S4D). These findings suggest that Olink may have quantified certain low-abundance proteins with higher precision, increasing the power to detect smaller differences. Still, MS uniquely identified many DAPs, generally higher in abundance, highlighting the methods' complementarity.

Finally, we examined the reproducibility of the DAPs in previous studies reporting sex-based differences in plasma protein levels measured with MS (one dataset)[33], Olink (two datasets)[34,35], and SomaScan (one dataset)[35]. The replication rate for each platform was calculated as the proportion of DAPs that were also present and statistically significant, with the same direction for the difference, in each published dataset. Replication rates varied substantially by study, ranging from 38–86% for MS, and 26–83% for Olink (Fig. S4E). Overall, replication rates were somewhat higher for MS, indicating that while Olink identified more DAPs, those identified by MS were more likely to replicate in independent datasets. This likely reflects differences in sensitivity and statistical power—MS may yield fewer but more confident DAPs, whereas Olink may detect subtle effects that require higher precision to replicate.

## Cross-platform correlation of protein levels

We estimated the quantitative agreement between HiRIEF LC-MS/MS and Olink Explore 3072 for each protein with Pearson and Spearman correlation coefficients (Fig. S5). While Pearson correlations were slightly higher on average, we interpreted the findings based on Spearman correlations due to their robustness to outliers and more conservative estimates.

For all proteins overlapping between MS and Olink ($N = 1129$), the median Spearman correlation between paired measurements was $\rho = 0.59$, with nearly two-thirds exhibiting moderate to strong correlations of

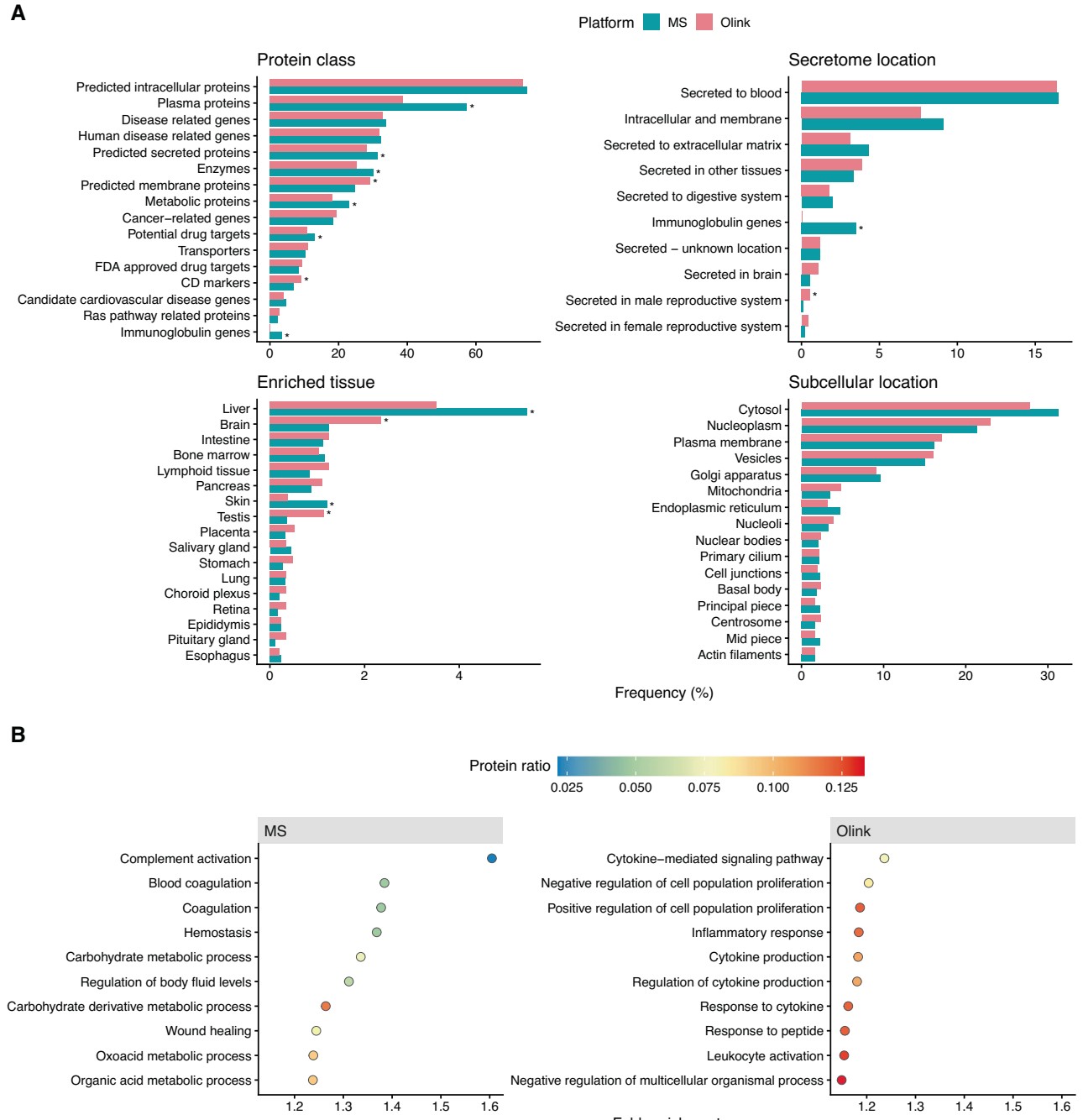

**Fig. 3 | Characterization of detected proteins. A** Comparison of the frequency of protein annotations from select Human Protein Atlas (HPA) categories among proteins detected with HiRIEF LC-MS/MS and Olink Explore 3072. The 15 most frequent annotations within each HPA category are shown for both platforms. Frequencies were calculated relative to the total number of proteins detected by each platform that were also found in the HPA. Asterisks indicate statistically significant differences in frequency between platforms (Fisher's exact test, false discovery rate <0.05). **B** Overrepresentation analysis of Gene Ontology (GO) Biological Processes among proteins detected with each technology. All proteins detected by MS and/or Olink were used as the background protein list ($N = 4362$). Points are colored by the proportion of input proteins (MS or Olink proteins) associated with each GO term.

$\rho \in [0.5, 1.0]$ (Fig. 5A, Supplementary Data 8). Several proteins had near-perfect agreement, with the highest correlations observed for MBL2, PZP, ANGPT1, MYL3, DPT, and SHMT1 ($\rho > 0.95$). In contrast, some proteins showed strong disagreement, with negative correlations, for example, PAXX, SRPK2, GLIPR1, IL10RB, ISM2, and LSM1 ($\rho < -0.40$). Proteins with very low cross-platform correlations generally had a large proportion of missing values or values < LOD (Fig. 5B).

Cross-platform correlations improved slightly after removing values < LOD and values with quality control (QC) warnings in the Olink

data ($N = 1064$, Fig. S6). On the subset of overlapping proteins with no missing values, values < LOD, or QC warnings ($N = 463$), the median correlation reached $\rho = 0.68$, with 81% of proteins exhibiting moderate to strong correlations between platforms (Fig. 5C).

Lastly, we calculated cross-platform Spearman correlations between different isoforms of the same protein (Supplementary Data 9). Matched by gene name, we identified 41 genes for which MS and Olink measured different isoforms, based on UniProt IDs. Of these, 21 had more than one isoform in the MS data. For most, the agreement with Olink was best for the

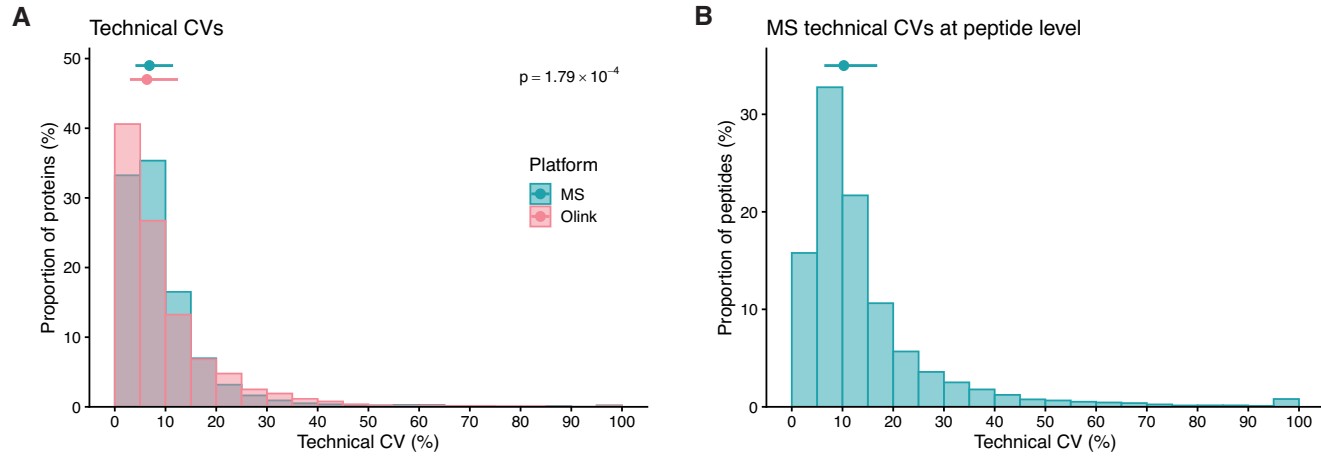

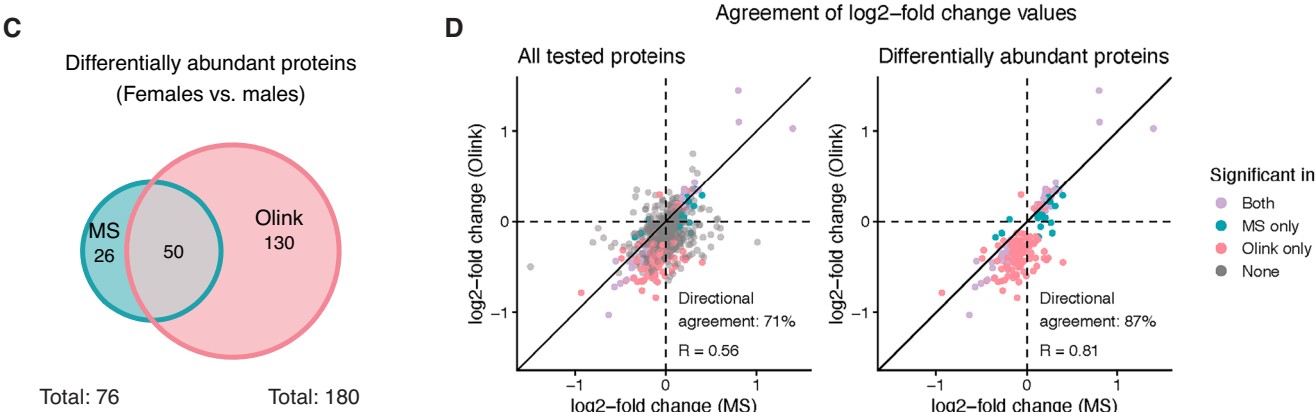

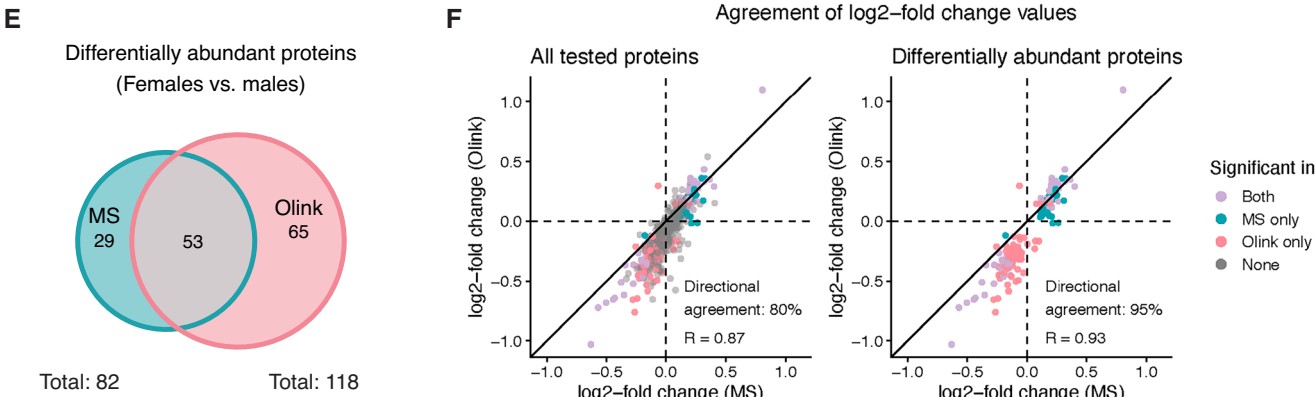

**Fig. 4 | Technical variation and agreement in differential abundance analyses.**
**A** Technical coefficients of variation (CVs) per protein for HiRIEF LC-MS/MS and Olink Explore 3072. Medians and interquartile ranges (IQR) are indicated with points and error bars. Differences in CVs between platforms were tested using a two-sided Wilcoxon rank-sum test. CVs were capped at 100%. **B** Technical CVs per peptide for MS. The median and IQR are indicated with a point and error bar. **C** Number of differentially abundant proteins (DAPs) between males and females by platform (two-sided Welch's $t$ test, false discovery rate < 0.05). The analysis included all overlapping proteins ($N = 1129$). **D** Agreement of log2-fold change values (calculated as female–male) between platforms for all overlapping proteins (left)

and for proteins identified as differentially abundant in at least one platform (right). Points are colored by which dataset(s) the protein was found to be differentially abundant in. The solid black line represents perfect agreement (slope = 1). Directional agreement was calculated as the percentage of proteins showing the same direction of change between females and males in both platforms. **E** Number of DAPs between males and females by platform, based on overlapping proteins with no missing values to ensure equal sample sizes in each test ($N = 569$). **F** Agreement of log2-fold change values between platforms for overlapping proteins with no missing values.

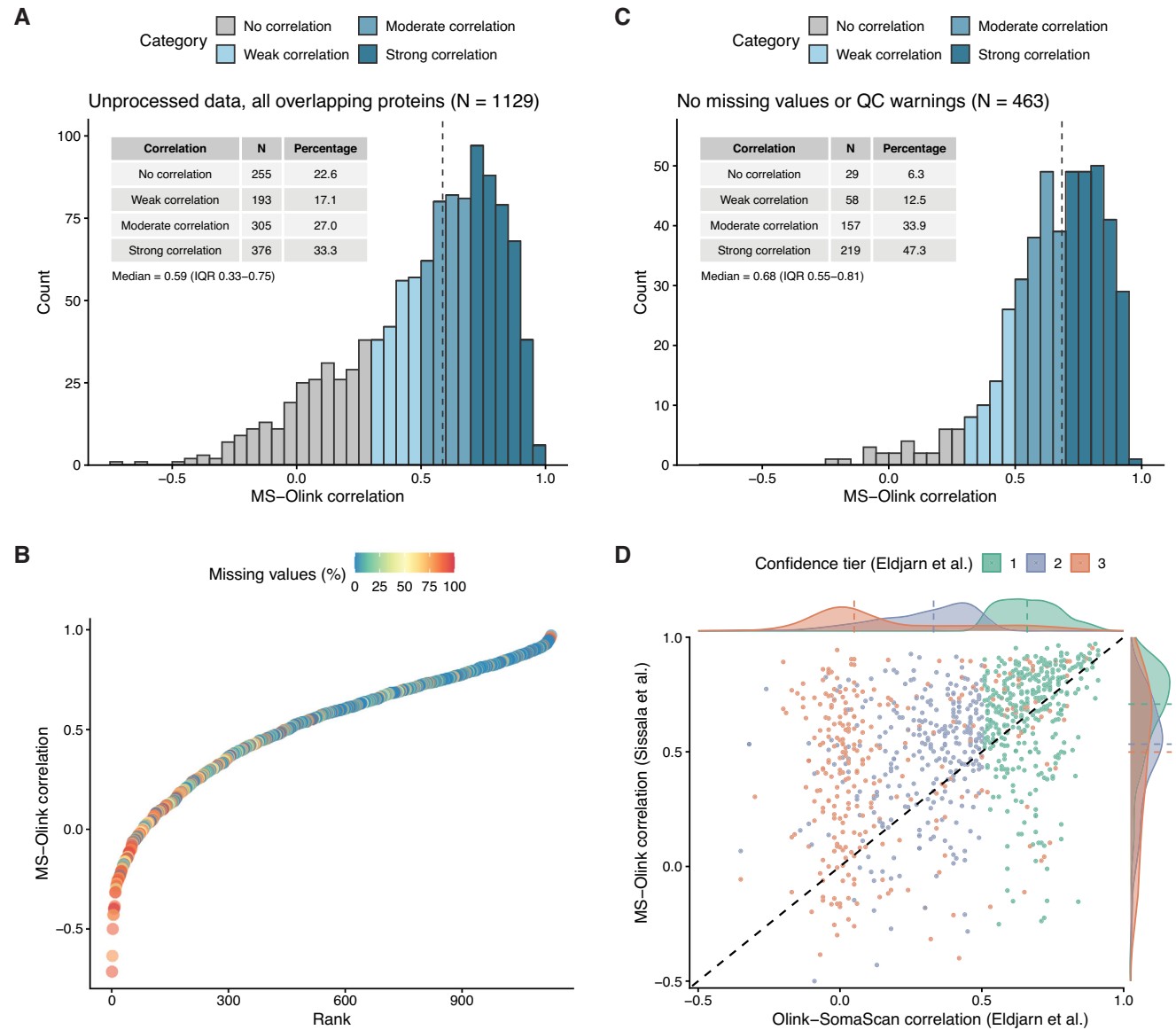

**Fig. 5 | Cross-platform correlation of protein levels. A** Histogram of Spearman correlations ($\rho$) between HiRIEF LC-MS/MS and Olink Explore 3072 protein measurements for all overlapping proteins ($N = 1129$). Missing values, values below the limit of detection (LOD) and quality control (QC) warnings were retained. Correlations were categorized as no correlation: $\rho \in [-1, 0.3)$; weak correlation: $\rho \in [0.3, 0.5)$; moderate correlation: $\rho \in [0.5, 0.7)$; and strong correlation: $\rho \in [0.7, 1.0]$. The dashed line indicates the median correlation. **B** Proteins ranked by MS-Olink correlation and colored by proportion of samples with missing values or values < LOD in MS or Olink data. **C** Same as A but restricted to overlapping proteins with no missing values, values < LOD, or QC warnings ($N = 463$). **D** MS-

Olink correlations from the present study versus Olink-SomaScan correlations from Eldjarn et al.[14] for matching proteins (matched by UniProt ID). Points are colored by the confidence tier assigned by Eldjarn et al. Tier 1 includes proteins with an Olink-SomaScan correlation >0.5 and a cis-pQTL detected by both platforms in Eldjarn et al., tier 2 includes proteins with a correlation of ≤0.5 and a cis-pQTL detected by both platforms, and tier 3 includes proteins with a cis-pQTL detected by only one or none of the platforms. The dashed line represents perfect agreement between studies (slope = 1). Density plots around the plot margins show the distribution of correlations by confidence tier, with dashed lines indicating the median in each tier.

canonical isoform. One notable exception was MASP1, for which the agreement was better for isoform 2 ($\rho = 0.57$) compared to the canonical isoform ($\rho = 0.04$), suggesting that the Olink assay may primarily target isoform 2.

### Comparison with previous studies

To assess the reproducibility of MS-Olink correlation estimates across studies, we compared the MS-Olink Spearman correlations from our dataset ($N = 1129$) with those reported in previous studies (see Methods and Supplementary Data 10)[25–28]. For comparability, we restricted the analysis to proteins measured in both our study and each external study. The previous studies reported varying levels of agreement between MS and Olink, with

median correlations ranging from $\rho = 0.27$ to $\rho = 0.56$ across overlapping proteins (Fig. S7, Supplementary Data 11). In comparison, our study consistently showed higher MS-Olink correlations for the corresponding proteins, with medians ranging from $\rho = 0.59$ to $\rho = 0.72$.

We then compared the MS-Olink correlations from our study to published cross-platform correlations involving the SomaScan platform (Olink-SomaScan and MS-SomaScan), again restricting the analysis to proteins measured in both our study and each external study[14–26]. In most cases, the median MS-Olink correlations from our study were higher than the corresponding Olink-SomaScan and MS-SomaScan correlations reported in previous studies (Figs. S8 and S9, Supplementary Data 12–13).

In a recent large-scale comparison of Olink and SomaScan platforms, Eldjarn et al.[14] categorized proteins into three confidence tiers according to the reliability of their measurements, with tier 1 representing the highest confidence, and tier 3 the lowest confidence. The classification was based on cross-platform correlations and the detection of protein quantitative trait loci (pQTLs) (see Methods). To provide additional orthogonal validation with MS, we examined our estimates of MS-Olink correlations in the context of these confidence tiers. Tier 1 proteins had a clearly higher median correlation between HiRIEF LC-MS/MS and Olink Explore 3072 ($\rho = 0.71$), compared to tier 2 ($\rho = 0.53$) and tier 3 ($\rho = 0.50$) proteins (Fig. 5D, Supplementary Data 14). This supports the idea that the presence of pQTLs on both platforms, along with a high cross-platform correlation, is indicative of more accurate protein quantification. The median correlation between HiRIEF LC-MS/MS and Olink Explore 3072 for all confidence tiers was higher than the corresponding Olink-SomaScan correlations (Fig. S10). However, the difference was most pronounced for tier 3 proteins, where the median HiRIEF-Olink correlation was $\rho = 0.50$, compared to $\rho = 0.05$ for Olink-SomaScan. These results suggest that the Olink assays for tier 3 proteins may be more accurate than previously indicated. Overall, our data provide orthogonal validation for the quantification accuracy of many Olink, and by extension SomaScan assays, in tier 1, and for a few assays in tier 3, based on strong cross-platform correlations in both studies ($\rho > 0.7$, Supplementary Data 14).

### Technical factors affecting cross-platform correlations

To explore how technical factors influenced the quantitative agreement between HiRIEF LC-MS/MS and Olink Explore 3072 measurements, we employed univariable linear regression with MS-Olink correlation as the dependent variable (Supplementary Data 15, Fig. 6A and Figs. S11–S14). Among all tested variables, the proportion of missing values in MS and the proportion of values < LOD in Olink explained the largest proportion of variance in the correlations (Fig. 6A). As expected, higher proportions of missing values were linked to lower MS-Olink correlations (Fig. 6B, C), likely reflecting noisier quantification, as proteins with more missing values had lower estimated concentrations, higher technical CVs, and median values closer to LOD in the Olink data (Fig. S15). Consequently, all these factors were also associated with weaker cross-platform correlations (Figs. S11 and S14).

Among MS-specific factors, the number of peptide-spectrum matches (PSMs) and unique peptides per protein (Fig. 6D), as well as sequence coverage, were the strongest predictors of higher MS-Olink correlations. As expected, these factors had an inverse association with missing values (Fig. S15). The correlations were lower for proteins with one median PSM or peptide, but still moderate on average (median $\rho = 0.42$, Fig. S16), suggesting that a low number of PSMs or peptides alone is not sufficient to deem a protein quantification unreliable. In contrast, proteins with high precursor mass errors had weak correlations with Olink (Fig. S13).

Among Olink-specific factors, a higher number of sample QC warnings per protein was associated with lower correlations (Fig. 6E), although few samples were affected (Fig. S17). Only 30 proteins had an assay QC warning, and these showed no difference in cross-platform correlations compared to the rest (Fig. 6E). The cross-platform correlation varied by Olink panel, likely driven by missing values (Fig. S14). Similarly, compared to version I panel proteins, correlations were somewhat lower for the version II panel proteins, which also had more missing values on average (Fig. S14).

### Protein characteristics affecting cross-platform correlations

Next, we explored potential similarities in protein properties and functional annotations among proteins with poor correlations between MS and Olink. We found no difference in cross-platform correlations based on protein mass, length, or the number of isoforms reported in the UniProt database (Fig. S18), and no enriched GO, MSigDB, KEGG, or Reactome gene sets. In contrast, enzymes, enzyme inhibitors, predicted secreted proteins, proteins secreted to the digestive system and the blood, as well as candidate cardiovascular disease genes from the HPA were enriched among proteins with

high cross-platform correlations (Fig. 6F, Supplementary Data 16). Most of these annotations were also overrepresented in the strong correlation group ($\rho \in [0.7, 1.0]$), while the no correlation group ($\rho \in [-1, 0.3)$) had an overrepresentation of proteins related to intermediate filaments, mainly keratins (Supplementary Data 17). These findings could be explained by the higher abundance of the proteins with strong MS-Olink correlations, while keratins could reflect sample contamination.

### Cross-platform correlations at the peptide level

Lastly, we examined cross-platform correlations between HiRIEF LC-MS/MS and Olink Explore 3072 at the peptide level to identify protein sequences or regions with differing agreement between platforms. Correlations were calculated between Olink protein measurements and corresponding MS peptide measurements, matched by gene name. To obtain more robust correlation estimates, we excluded peptides quantified in fewer than 15 samples with MS, resulting in a dataset of 13,856 peptides, mapping to 822 genes and 847 unique UniProt IDs (Supplementary Data 18). To enhance the accessibility of the results, we developed PeptAffinity, a publicly available interactive R Shiny app (https://peptaffinity.serve.scilifelab.se/app/peptaffinity). PeptAffinity allows users to visualize peptides quantified by MS on the protein sequence, along with their correlation with the corresponding Olink assay. Furthermore, for visualizing the correlations in 3D, we annotated the peptide correlations on protein structures, as predicted by AlphaFold[36,37]. Below, we provide a few representative examples to illustrate the utility of PeptAffinity in exploring cross-platform correlations: Protein AMBP (*AMBP*), CD99 antigen (*CD99*), Hypoxia upregulated protein 1 (*HYOU1*), and Mannan-binding lectin serine protease 1 (*MASP1*). These proteins exhibited substantial variation in peptide-Olink correlations across different regions of their respective protein sequences, consistent with differential measurement of specific isoforms with MS and Olink.

The *AMBP* gene encodes a precursor protein that is cleaved into two distinct functional products: α1-microglobulin and inter-α-trypsin inhibitor (IαI) light chain. Notably, peptides mapping to the α1-Microglobulin region showed stronger correlations with the AMBP Olink assay (median $\rho = 0.53$) than those mapping to the IαI Light Chain region (median $\rho = 0.07$) (Fig. 7A, B and Fig. S19), suggesting that the Olink assay primarily measures the α1-microglobulin proteoform.

CD99 is a membrane glycoprotein that consists of a cytoplasmic (intracellular), transmembrane, and extracellular domain. The protein exists as two different proteoforms: a long form that contains both intra- and extracellular domains, and a short form that has a truncated intracellular region. Peptides mapping to the intracellular domain had lower correlations with the CD99 Olink assay (median $\rho = 0.12$) compared to the extracellular domain (median $\rho = 0.64$), indicating that measured CD99 plasma protein levels primarily reflect the short isoform (Figs. 7C and S19).

Similarly, for HYOU1, we observed two regions with differing cross-platform correlations. One region, shared between isoforms 1 and 2, demonstrated poor agreement between MS and Olink measurements (median $\rho = 0.21$), whereas the other, unique to isoform 1, exhibited moderate agreement (median $\rho = 0.64$) (Figs. 7D and S19), indicating that the Olink assay mainly targets isoform 1.

Finally, we confirmed that MASP1, which had multiple isoforms in the MS data and the strongest correlation with Olink at the protein level for isoform 2, exhibited higher cross-platform correlations for peptides mapping to regions unique to isoform 2 than peptides mapping to other regions (Figs. 7E and S19). These findings further suggest that Olink's antibodies may be binding sequences specific to isoform 2 of MASP1.

In summary, these examples illustrate how peptide-level analysis by MS can provide a more detailed view of protein quantification across platforms than protein-level analysis alone, clarifying discrepancies and revealing potential differences in proteoform measurements.

### Discussion

In this study, we present a thorough technical comparison of peptide fractionation-based global MS proteomics (HiRIEF LC-MS/MS) and the

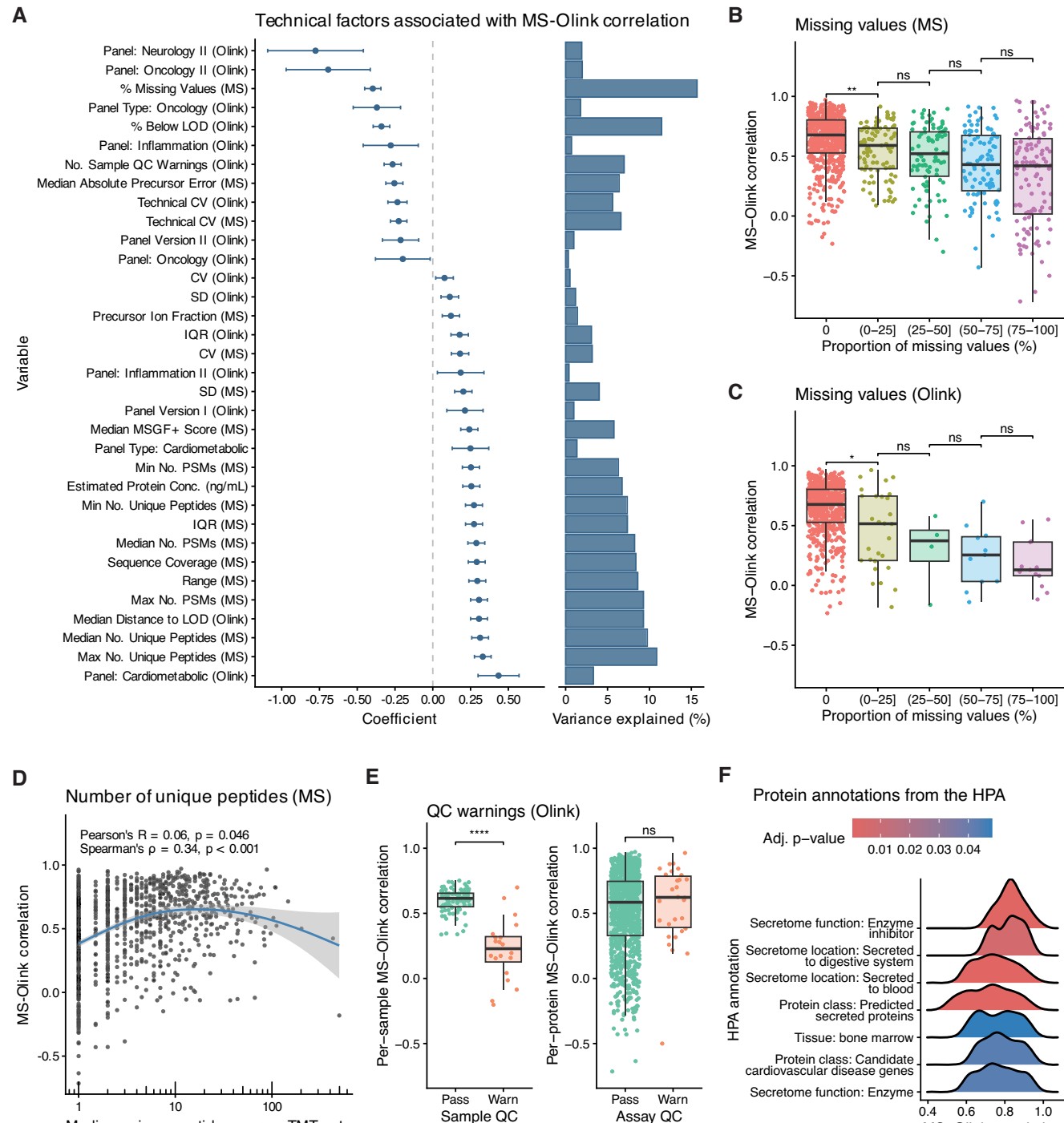

**Fig. 6 | Effect of technical factors and protein characteristics on cross-platform correlations. A** Technical factors influencing the agreement (Spearman correlation) between HiRIEF LC-MS/MS and Olink Explore 3072 protein measurements (linear regression, false discovery rate (FDR) < 0.05). Positive coefficients indicate associations with higher MS-Olink correlations. Error bars represent 95% confidence intervals. The percentage of variance explained by each factor is shown on the right, expressed as adjusted $R^2$. **B** MS-Olink correlation by the proportion of missing values per protein in MS. Proteins with any missing values in the Olink data were excluded. *P* values were determined using a two-sided Wilcoxon rank-sum test and adjusted for multiple testing with the FDR method. ns = not significant, **$p < 0.01$. **C** MS-Olink correlation by the proportion of missing values per protein in Olink. Proteins

with any missing values in the MS data were excluded. *P* values were determined as in **B**. ns not significant, *$p < 0.05$. **D** MS-Olink correlation versus median number of unique peptides used for quantification across tandem mass tag (TMT) sets with MS. The *x* axis is on a $\log_{10}$ scale. **E** Left: comparison of per-sample MS-Olink correlations of normalized protein expression (NPX) values with ("Warn") versus without ("Pass") a sample quality control (QC) warning in the Olink data. Right: comparison of the MS-Olink correlation of proteins with ("Warn") and without ("Pass") an assay QC warning in the Olink data. *P* values were determined as in **B**, **C**. ns not significant, ****$p < 0.0001$. **F** Human Protein Atlas (HPA) annotations enriched among MS-Olink correlations at 5% FDR. Each row shows the distribution of MS-Olink correlations for proteins associated with a specific HPA annotation.

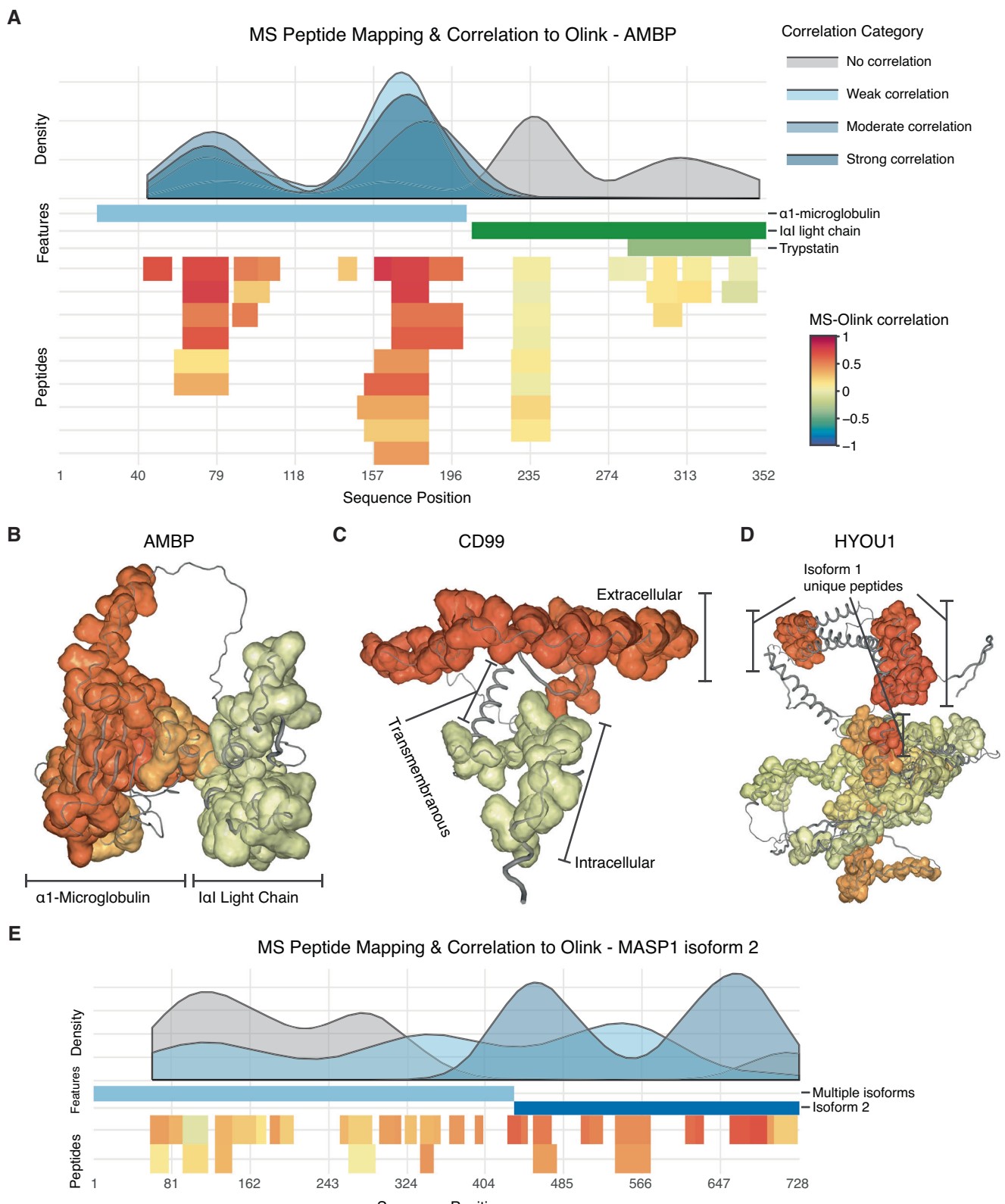

**Fig. 7 | Cross-platform correlations at the peptide level for AMBP, CD99, HYOU1, and MASP1.** Visualization of cross-platform correlations between HiRIEF LC-MS/MS peptides and Olink Explore 3072 protein assays for **A** sequence of AMBP, **B** structure of AMBP, **C** structure of CD99, **D** structure of HYOU1 (isoform 1), and **E** sequence of MASP1 (isoform 2). In the sequence plots, MS peptides and select sequence features (y axis) are plotted along the protein sequence (x axis). Each horizontal bar represents either a sequence feature (middle part of each plot) or a peptide (bottom part of each plot). Peptides are shown in multiple rows to avoid overlaps and are colored by their Spearman correlation ($\rho$) with the corresponding Olink assay. At the top of each plot, a density graph illustrates the distribution of peptide-Olink correlations across the protein sequence, divided into categories of no correlation: $\rho \in [-1, 0.3)$; weak correlation: $\rho \in [0.3, 0.5)$; moderate correlation: $\rho \in [0.5, 0.7)$; and strong correlation: $\rho \in [0.7, 1.0]$. In the structure plots, correlations between MS peptide levels and the matching Olink assay are mapped onto protein structures predicted with AlphaFold. The color legends for the correlation categories and the MS-Olink correlations (shown in **A**) apply to all panels.

Olink Explore 3072 platform, based on an in-depth analysis of the plasma proteome involving 4362 proteins in total and 1129 proteins overlapping between methods—more than in most previous studies comparing MS and Olink's PEAs[26–28]. We show that these platforms exhibit complementary proteome coverage, high precision, good concordance and complementarity in estimating sex differences in protein levels, and largely good, albeit variable, quantitative agreement in protein levels. Consequently, we provide orthogonal validation for a large proportion of the Olink Explore 3072 assays. Furthermore, we identify technical factors and protein properties influencing the quantitative agreement and compare estimates of cross-platform agreement across previous studies. Finally, we demonstrate how a peptide-level analysis of cross-platform correlations can reveal insights into differences in proteoform measurement.

Our findings demonstrate the complementarity of global MS and Olink Explore 3072 in providing a more comprehensive analysis of the plasma proteome than either method alone, in line with previous reports[27,28]. This complementarity was evident in the analysis of sex differences in plasma protein levels. A greater number of DAPs were identified when combining both platforms, with DAPs unique to each platform originating from different concentration ranges. While only a subset of DAPs reached statistical significance with both methods, there was strong concordance in the direction and magnitude of the estimated differences. This demonstrates that since discrepancies in statistical significance can arise from differences in precision and data completeness, it is of value to examine trends across platforms regardless of statistical significance. Overall, combining these technologies offers broad proteome coverage across a wide concentration range and distinct biological processes, increasing the potential for biologically and clinically relevant discoveries.

As combining platforms is not always feasible, the choice of method should be guided by their strengths and limitations, as well as the specific goals of the study. The differences in proteome coverage suggest that each platform may offer distinct advantages depending on disease context. For example, the relatively higher coverage of classical plasma proteins and metabolic proteins observed for MS may be advantageous for investigating systemic effects on the plasma proteome. In contrast, Olink offers an advantage in analyzing low-abundance, tissue-leakage, and signaling proteins. The Olink Explore platform currently offers higher sensitivity and throughput than MS, enabling large-scale studies with high proteome coverage, though limited to pre-defined targets. In contrast, the untargeted nature of MS provides a significant advantage for discovery-focused studies, allowing for a general characterization of the plasma proteome across conditions and the detection of novel proteins and post-translational modifications (PTMs). This advantage will likely become more pronounced as sensitivity, proteome coverage, and speed continue to improve. In particular, improvements in enrichment methods and MS instrumentation have enabled higher sample throughput while maintaining or increasing profiling depth[38–42]. However, recent multi-platform comparisons of plasma proteomics technologies have shown that the precision and agreement of these developing MS workflows with Olink and SomaScan are not necessarily improved[25,43]. Similarly, advances in the Olink technology have increased proteome coverage and sample throughput, albeit with a greater proportion of values < LOD and lower agreement with other proteomic methods for the Olink Explore HT platform[21,25,43].

The cross-platform correlation analysis revealed comparable but slightly higher correlations between MS and Olink than reported in previous studies[25–28]. Several technical factors were associated with weaker cross-platform agreement, including higher proportions of missing values, lower estimated protein concentrations, and higher technical CVs; fewer PSMs or peptides for MS data; and sample QC warnings and median NPX values closer to the LOD for Olink data. Yet, several proteins with seemingly high-quality quantification on both platforms had poor agreement between platforms. This could be explained by a multitude of factors: isoforms, sample handling or preparation, antibody cross-reactivity, PTMs, epitope effects, and many more. Previous studies have made significant efforts to

validate antibody specificity and selectivity using MS[44–46], and similar work is needed to evaluate the affinity binders used in commercial plasma proteomics platforms. Our data provide orthogonal validation for the accuracy and specificity of at least one third of the Olink Explore 3072 assays, based on strong agreement with HiRIEF LC-MS/MS measurements. This represents one important pillar of affinity binder validation[47].

In addition to quantitative agreement, others have explored the utility of incorporating associations with genetic variation—the detection of pQTLs—to assess the accuracy of affinity-based assays[14,15,18]. However, based on our comparison with results from Eldjarn et al.[14], we show that the pQTL approach is not without limitations, as proteins with pQTLs on both SomaScan and Olink platforms did not necessarily have a strong correlation between each other or with MS. Conversely, for several proteins, the accuracy of the affinity-based assays was supported by our MS data despite a lack of pQTLs. This highlights the need to validate pQTLs detected with affinity-based methods through cross-platform comparisons with MS, which in turn will require MS analyses with large sample sizes.

Through a peptide-level analysis of cross-platform correlations, we identified differential detection of proteoforms produced by proteolytic cleavage or alternative splicing. The presented examples illustrate how aggregating peptide signals into a single protein can mask meaningful biological variation, demonstrating the importance of shifting from a gene- or protein-centric to a proteoform-centric analysis of the proteome. Since different proteoforms can have different cellular functions, localizations, or roles in disease, measuring them precisely can advance biomarker discovery and drug development[48]. For example, CD99 is a relevant diagnostic biomarker and potential therapeutic target in certain sarcomas and hematological malignancies[49]. However, its isoforms have shown distinct and sometimes opposing effects on tumor progression[49]. Measuring the isoforms as a single entity could obscure these differences, leaving potential clinically significant discoveries undetected. In this context, MS proteomics has a unique advantage in distinguishing between proteoforms, while capturing these signals with affinity proteomics would require further resolving the specific epitopes and isoforms targeted by the binders.

Some limitations of this study should be considered. First, the CV calculations were based on a small number of duplicate samples, introducing uncertainty in the estimates, and intra- and inter-assay CVs could not be calculated for both technologies. Second, the results may not be fully generalizable to other MS workflows or Olink platforms. Differences in cohort characteristics, sample type, sample preparation protocols, assay versions, instrumentation, and data processing could contribute to variability between studies. Third, we aimed to evaluate the overall agreement between our in-house plasma proteomics protocol and Olink, but from an MS perspective, future studies would benefit from exploring how factors such as depletion, fractionation, multiplexing, MS3 quantification, data acquisition strategy, and data processing affect quantitative agreement with affinity-based methods. Lastly, since data completeness affects both the quantitative agreement between platforms and the statistical significance of biological findings, future work would benefit from assessing the comparability of MS and Olink results using imputed data. However, as the outcome of such an analysis would depend heavily on the chosen imputation method, it would require a thorough evaluation of different imputation strategies. Given the high proportion of missing values in global MS data and the increasing number of values below LOD in Olink data, evaluating imputation methods on different types of plasma proteomics data remains an important future direction.

In conclusion, this study illustrates how technical differences between peptide fractionation-based MS and Olink Explore 3072 influence the reproducibility of findings in plasma proteome profiling. Our analysis, encompassing a large number of overlapping proteins, a thorough investigation into factors affecting platform performance, and quantitative agreement at both the protein and peptide level, demonstrates the complementary strengths of MS and affinity-based proteomics and provides insights for platform selection and study design. Ultimately, our results highlight the added value of combining these platforms for a more

comprehensive and reliable profiling of the plasma proteome, enabling broader characterization of different diseases and more robust biomarker discovery.

## Materials and methods
### Ethical approval
This study was approved by the regional ethical review board in Stockholm, Sweden (EPN: ref no 2014/1290–32) and conducted in accordance with the Declaration of Helsinki. All participants provided written informed consent.

### Study design and sample collection
The present study cohort consists of a retrospectively selected subset of the PEX-LC cohort, which has been described previously[50]. Briefly, plasma samples were collected from patients referred to the Karolinska University Hospital (KUH) in Stockholm, Sweden, for investigation of suspected lung cancer between September 2014 and November 2015. The plasma samples were collected during the participants' first visit to KUH, before diagnosis and treatment. Blood was drawn into EDTA tubes, centrifuged at $2500 \times g$ at RT for 10 min, and the resulting plasma samples were biobanked and stored at $-80\,°C$. The present study cohort includes 114 patients, with an equal number of patients diagnosed with either lung cancer or no cancer, i.e., other benign lung conditions. Samples from all 114 patients were analyzed using HiRIEF LC-MS/MS, and a subset of 88 plasma samples with Olink Explore 3072. For the MS analysis, six samples were run in duplicate (aliquoted at the start of sample preparation), resulting in a total of 120 samples.

### Plasma proteome profiling
#### MS-based proteome profiling
**Plasma depletion and in-solution digestion**. To reduce sample complexity and increase the number of protein identifications, the 14 most abundant plasma proteins were depleted from the samples using High Select Top14 Abundant Protein Depletion Mini Spin Columns (Thermo Scientific). 10 µL of plasma was applied to each column, the columns were incubated at room temperature for 20 min with gentle end-over-end mixing, and depleted flowthroughs were obtained through centrifugation. The sample buffer was exchanged to 50 mM HEPES (pH 7.6) using 5 kDa spin concentrators (5 K MWCO, 4 mL, Agilent Technologies, 5185–5987) by centrifuging three times at 5000 rpm for 30 min. Protein concentration was measured using the Micro BCA Protein Assay Kit (Thermo Scientific, 23235) to estimate the total protein amount per sample.

Next, proteins were digested into peptides using lysC and trypsin (sequencing grade modified, Pierce) following a previously described in-solution digestion protocol[51]. In brief, 40 µg of protein from each sample was alkylated with 8 mM chloroacetamide. 100 µL of lysC buffer (0.5 M Urea, 50 mM HEPES, pH 7.6 and 1:50 enzyme-to-protein ratio) was added, and the samples were incubated overnight. The same procedure was repeated for trypsin; 100 µL of trypsin buffer (50 mM HEPES, pH 7.6, 1:50 enzyme-to-protein ratio) was added, and the mixtures were incubated overnight. Finally, the samples were dried in a SpeedVac and resuspended in 50 µL TEAB pH 8.5 to a final concentration of 100 mM.

**TMT labeling**. 40 µg of peptides from each sample was labeled with isobaric TMTs (TMTpro 16plex Label Reagent Set, Thermo Scientific) according to the manufacturer's protocol. A total of 120 samples (114 distinct plasma samples and six pairs of technical replicates) were labeled with eight sets of TMTpro 16plex, with one internal standard per set. The master pool of internal standards was made by pooling a small amount of protein from each sample, and the master pool was then split into eight internal standards of 40 µg of protein each. The TMT labeling scheme is shown in Table S1.

The TMT labeling efficiency was determined by LC-MS/MS prior to pooling of the samples. For this, 1 µL of each sample of a TMT set was mixed, dried down, and resuspended in 10 µL of mobile phase A. Approximately 2 µg was injected into the LC-MS/MS system and analyzed with a 3 h gradient. After confirming a labeling efficiency >95%, samples of the same TMT set were pooled. The eight resulting TMT pools were purified through solid phase extraction using SPE strata-X-C columns (Phenomenex), and purified samples were dried in a SpeedVac.

**High-resolution isoelectric focusing**. To further reduce sample complexity, pooled samples were pre-fractionated using HiRIEF, following a previously described protocol[52]. Briefly, peptides were separated by their isoelectric point through immobilized pH gradient isoelectric focusing (IPG-IEF) on gel strips with a 3–10 pH gradient. After IEF, each gel strip was split into 72 fractions, and proteins from each fraction were eluted and transferred to a 96-well microtiter plate using a liquid-handling robot (GE Healthcare prototype). Finally, the fractionated samples were dried in a SpeedVac and stored at $-20\,°C$ until analysis with LC-MS/MS.

**Liquid chromatography–MS analysis**. Online LC-MS was performed as previously described[52] using a Dionex UltiMate™ 3000 RSLCnano System coupled to a Q-Exactive-HF mass spectrometer (Thermo Scientific). The contents of each plate well were dissolved in 20 µL of solvent A and 10 µL was injected. Samples were trapped on a C18 guard-desalting column (Acclaim PepMap 100, 75 µm × 2 cm, nanoViper, C18, 5 µm, 100 Å) and separated on a 50 cm long C18 column (Easy spray PepMap RSLC, C18, 2 µm, 100 Å, 75 µm × 50 cm). The nano capillary solvent A consisted of 94.9% water, 5% DMSO, and 0.1% formic acid, and solvent B consisted of 5% water, 5% DMSO, 89.9% acetonitrile, and 0.1% formic acid. At a constant flow of 0.25 µL/min, the curved gradient went from 6–10% solvent B up to 40% solvent B in each fraction in a dynamic range of gradient length (see Table S2), followed by a steep increase to 100% solvent B in 5 min.

FTMS (Fourier transform MS) master scans with 60,000 resolution and mass range 300–1500 m/z were followed by data-dependent MS/MS with a resolution of 30,000 on the top 5 ions using higher energy collision dissociation at 30% normalized collision energy. Precursors were isolated with a 2 m/z window. Automatic gain control targets were $1^6$ for MS1 and $1^5$ for MS2. Maximum injection times were 100 ms for MS1 and 400 ms for MS2. The entire duty cycle lasted ~2.5 s. Dynamic exclusion was used with 30 s duration. Precursors with unassigned charge state or charge state 1 were excluded. An underfill ratio of 1% was used.

**Protein identification and quantification**. Orbitrap raw MS/MS files were converted to mzML format using msConvert from the ProteoWizard tool suite[53]. Spectra were searched using the ddamsproteomics Nextflow (v22.10.5)[54] pipeline (https://github.com/lehtiolab/ddamsproteomics, v2.11), which runs MSGF+ (v2020.03.14)[55] and Percolator (v3.04.0)[56] for peptide identification. All searches were performed against a database of all human proteins from the UniProtKB/Swiss-Prot release of May 2022. MSGF+ settings included precursor mass tolerance of 10 ppm, fully tryptic peptides, a maximum peptide length of 50 amino acids and a maximum charge of 6. Fixed modifications included carbamidomethylation on cysteine residues and TMTpro 16plex on lysine residues and peptide N-termini. A variable modification was used for oxidation on methionine residues. PSMs found at 1% FDR were used to infer protein identities.

TMTpro 16plex reporter ions were quantified using OpenMS project's IsobaricAnalyzer (v2.5.0)[57]. Relative quantification was calculated on the peptide and protein levels based on PSMs mapping to only one protein group (UniProt ID) and with 1% FDR. PSMs with missing values in any channel within a TMT set were excluded. Relative quantification values were calculated for each TMT channel as the median of PSM ratios (channel/internal standard). To obtain these PSM ratios, PSM intensities were log2-transformed, and the transformed PSM intensities of the internal standard were subtracted from the transformed PSM intensities of the channel. Peptide/protein quantification values were then normalized by subtracting the median of the channel from each value. Protein FDRs were calculated using the picked-FDR method using UniProt IDs as protein groups and limited to 1% FDR.

### Antibody-based proteome profiling
The samples were analyzed using the Olink Proteomics PEA Explore 3072 at SciLifeLab Affinity Proteomics unit at Uppsala University and the National

Genomics Infrastructure Uppsala. The detailed protocol for PEA Explore has been previously described by Wik and colleagues[13].

In summary, Olink's PEA Explore technology utilizes pairs of antibodies conjugated with single-stranded DNA oligonucleotide reporter molecules, known as probes, which bind to their respective targets if present in the sample. When both probes in a pair bind to their target in proximity, double-stranded DNA amplicons are generated. The Explore 3072 assay comprises eight distinct 384-plex panels targeting inflammation, oncology, cardiometabolic, and neurology proteins, covering a total of 2923 human proteins. Four of these are version I panels (e.g., Inflammation), which were part of the earlier Olink Explore 1536 platform, while the remaining four are version II panels (e.g., Inflammation II), added to the Olink Explore 3072 platform.

Following the initial probe-based immune reaction step in the PEA Explore workflow, the amplicons were extended and amplified in a two-step process, with individual sample index sequences added during the second step. After pooling the samples, the libraries were prepared and sequenced on a NovaSeq 6000 instrument (Illumina, San Diego, CA, USA). The raw BCL files were converted into count files, which were then translated into NPX values through a QC and normalization process incorporating internal and external controls, as specified by the manufacturer. In this process, QC is performed for each assay (protein) measured in each sample and for each assay overall. If the QC for an assay in a specific sample fails, the measurement receives a sample QC warning. If the overall assay QC fails, the assay receives an assay QC warning (across all samples).

The NPX data are presented on a log2 scale, where an increase of one NPX unit corresponds to a doubling of the protein content. A high NPX value indicates a high protein concentration. Each measured protein has a LOD determined at run time based on negative controls. Values < LOD and QC warnings were retained in all analyses, unless stated otherwise.

## Statistical analysis

**Proteome coverage.** The reference plasma proteome was compiled from proteins listed in the HPPP (PeptideAtlas build 2023-04, www.peptideatlas.org)[29], proteins with an estimated blood concentration in the HPA (v24, www.proteinatlas.org)[30,31], and proteins classified as secreted to blood in the HPA, resulting in a reference set of 4889 unique proteins based on UniProt IDs. Proteins were matched between MS, Olink, and HPPP datasets using UniProt IDs. For the HPA data, which lacked UniProt IDs for some proteins, matches with the MS and Olink datasets were identified primarily based on UniProt IDs and secondarily based on gene names. Proteins are referred to throughout the text with their corresponding gene names.

Estimated blood protein concentrations were obtained from the HPA[31] and converted to ng/mL. The HPA reports concentrations derived from both immunoassay data from the literature and MS data from the PeptideAtlas[29]. MS-based concentrations were used when available. For proteins lacking MS-based concentrations, immunoassay-based concentrations were used, if available.

**Comparison of protein annotations.** The frequency of specific HPA annotations was calculated among all proteins detected in at least one sample by MS or Olink. Overrepresentation in either platform was tested using a hypergeometric test, with all proteins detected by MS and/or Olink used as the background ($N = 4362$). P values were adjusted for multiple testing using the FDR method, with a significance threshold of FDR < 0.05. The "Enriched tissue" category refers to proteins annotated as "Tissue enriched" in the Tissue section of the HPA[30], meaning their mRNA expression was at least four-fold higher in a specific tissue compared to all other tissues.

ORA of GO Biological Processes between platforms was performed using the compareCluster function in the clusterProfiler (version 4.14.4)[58] R package. All proteins detected with MS and/or Olink were used as the background protein list ($N = 4362$). Fold enrichment for GO terms was calculated as the ratio of the frequency of the input proteins (GeneRatio) to

the frequency of the background proteins (BgRatio) found in the respective GO term protein list, based on UniProt IDs. For Olink assays with multiple UniProt IDs, the first one was used. P values were adjusted for multiple testing using the FDR method, with a significance threshold of FDR < 0.05.

The coverage of FDA-approved plasma protein biomarkers was based on a list compiled by Anderson[32]. Proteins were matched between this list and the MS and Olink datasets based on UniProt IDs. Ten markers with no UniProt ID were excluded from the analysis.

**Technical CVs.** Technical CVs were calculated using the CV formula for data on a log$_2$-scale[59]:

$$CV = 100\% * \sqrt{e^{(\ln(2)*SD)^2} - 1}$$

CVs were calculated on up to six duplicate samples for MS, and the final technical CV was calculated as the mean of the individual duplicate CVs. For Olink, the technical CVs were calculated on one duplicate sample, the Olink Sample Control, which is a standard pooled plasma sample. CVs were capped at 100%.

**Differential abundance analysis.** DAAs between females ($N = 37$) and males ($N = 51$) were performed using a two-sided Welch's $t$ test, with males as the reference group. P values were adjusted for multiple testing per platform using the FDR method, with a significance threshold of FDR < 0.05. The directional agreement of the log2-fold change values was calculated as the percentage of proteins showing the same direction of change (positive or negative) in both platforms.

For the calculation of replication rates for MS and Olink DAPs, DAA results reporting differences in plasma protein levels between males and females were obtained from the supplementary materials of three previous publications[33–35]. The replication rates were calculated, out of the DAPs tested in both the present study and each previous study, as the proportion of DAPs with a statistically significant difference in the same direction in both studies.

**Cross-platform correlation analyses.** Correlations between MS and Olink measurements of matched proteins were calculated using both Pearson and Spearman correlation coefficients, and all correlations were presented without filtering for statistical significance. Proteins with fewer than eight overlapping data points were excluded from the analyses. Correlations were calculated on the full dataset of all overlapping proteins ($N = 1129$), a cleaned dataset where values < LOD and QC warnings in the Olink data were set to missing ($N = 1064$), and a dataset of overlapping proteins with no missing values or QC warnings in either platform ($N = 463$). The cross-platform correlations were divided into categories of no correlation: $\rho \in [-1, 0.3)$; weak correlation: $\rho \in [0.3, 0.5)$; moderate correlation: $\rho \in [0.5, 0.7)$; and strong correlation: $\rho \in [0.7, 1.0]$.

For the peptide-level analysis, MS peptides were matched to Olink assays based on gene name (i.e., Olink assay name), and correlations were calculated using Spearman's rank correlation coefficient. Peptides quantified in fewer than 15 samples, and genes with fewer than two peptides were excluded from the analysis. Information on the sequence positions of different isoforms and cleavage products of AMBP, CD99, HYOU1, and MASP1 was obtained manually from UniProt (release 2025_01, https://www.uniprot.org/). The fasta sequences of proteins included in the PeptAffinity R Shiny app were downloaded from UniProt (release 2022_05).

Associations between MS-Olink correlations and technical factors were assessed using univariable linear regression models with the MS-Olink Spearman correlation as the dependent variable. P values were adjusted using the FDR method, with a significance level of FDR < 0.05. Information on protein mass, length, and number of isoforms was obtained from UniProt release 2023_03, and protein concentration data were obtained from the HPA as described above. All other technical factors analyzed were obtained or calculated from the MS and Olink data files.

**Article**

GSEAs of GO terms, HPA annotations, and MSigDB, KEGG, and Reactome gene sets were performed using the clusterProfiler R package, with a ranked list of the Spearman correlations between MS and Olink measurements of all overlapping proteins as the input ($N = 1129$). $P$ values were adjusted for multiple testing using the FDR method, with a significance threshold of FDR < 0.05.

ORAs of GO terms, HPA annotations, and MSigDB, KEGG, and Reactome gene sets among proteins in the low correlation ($\rho < 0.3$) and strong correlation ($\rho \geq 0.7$) categories were performed as described above for the comparison of GO terms between platforms. All overlapping proteins ($N = 1129$) were used as the background.

**Comparison with previous studies.** The previous studies included in the comparison of cross-platform correlations are summarized in Supplementary Data 10. Cross-platform correlations were obtained from the supplementary materials of all publications except Petrera et al.[27], for which MS and Olink data files were downloaded, and Spearman correlations were calculated between the DDA-MS and Olink measurements of all overlapping proteins matched by UniProt IDs. Proteins were matched between studies primarily by UniProt IDs or by gene name if UniProt IDs were not provided. For proteins measured by multiple affinity reagents within one platform, the cross-platform correlation was calculated as the median correlation of all matched reagent pairs. Several of the studies comparing Olink and SomaScan provided cross-platform correlations for both normalized and non-normalized SomaScan data. For these studies, the correlations calculated from normalized data were used.

For the comparison of correlations by confidence tiers defined in Eldjarn et al.[14], data were obtained from Supplementary Table 29 of the original publication. The authors defined the confidence tiers as follows: tier 1, the highest confidence tier, included proteins with an Olink-SomaScan correlation >0.5 and cis-pQTLs detected on both platforms; tier 2 included proteins with a correlation of ≤0.5 and a cis-pQTL detected on both platforms; and tier 3 included proteins with a cis-pQTL identified on only one or neither platform.

**Software.** All statistical analyses and data visualizations were performed in R (version 4.4.2)[60]. Information on R packages and versions used is provided in the code repository on GitHub.

### Reporting summary
Further information on research design is available in the Nature Portfolio Reporting Summary linked to this article.

## Data availability
The HiRIEF LC-MS/MS data have been deposited to the ProteomeXchange Consortium via the PRIDE partner repository with the dataset identifier PXD061144. The Olink Explore 3072 data have been deposited in the PRIDE repository with the dataset identifier PAD000006. Individual-level personal data are not publicly available as they contain information that could compromise participants' privacy. Data used for the PeptAffinity R shiny app can be found in the code repository on GitHub. All other data supporting the findings of this study are available within the paper and its supplementary information files. Figure source data are provided as Supplementary Data 19.

## Code availability
The R code used for the analyses is deposited at https://github.com/noorasissala/MS-Olink-comparison[61]. The code for the PeptAffinity R Shiny app is deposited at https://github.com/isabelle-leo/PeptAffinity[62]. The ddamsproteomics Nextflow pipeline for DDA-MS data searches is available at https://github.com/lehtiolab/ddamsproteomics.

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

## Acknowledgements

We acknowledge support from the Global Proteomics and Proteogenomics Unit and the Affinity Proteomics Unit at the Science for Life Laboratory, as well as the National Genomics Infrastructure Uppsala. This project was funded by the Sjöberg Foundation (ref no 2022-01-11:7, to L.E.E.), the Swedish Research Council (ref no 2022-01176, to L.E.E.), the Cancer Society in Stockholm (ref. no 211073, to L.E.E.), and the SciLifeLab Technology Development Grant 2022 (to M.P.). I.R.L. was supported by Grant Number T32 AI155387 from NIAID/NIH. The contents of this publication are solely the responsibility of the authors and do not necessarily represent the official views of the NIAID or NIH.

## Author contributions

Conceptualization: M.P., M.Å., C.F., N.S. Data curation: J.F. Formal analysis: N.S. and I.R.L. Funding acquisition: M.P., M.Å., C.F., J.L., L.E.E. Methodology: M.P., M.Å., C.F., N.S., I.R.L., H.B. Resources: M.P., M.Å., C.F., L.E.E., J.L. Software: I.R.L., N.S. Investigation: X.C., M.Å. Visualization: N.S., I.R.L. Supervision: M.P., J.L., H.B., L.E.E. Writing—original draft: N.S., M.P. Writing—review & editing: N.S., H.B., I.R.L., X.C., J.F., L.E.E., J.L., C.F., M.Å. and M.P.

## Funding

## Competing interests

The authors declare no competing interests.
