## [Transparent Peer Review file · Communications Chemistry]

Comparative evaluation of Olink Explore 3072 and mass spectrometry with peptide fractionation for plasma proteomics

Corresponding Author: Ms Noora Sissala

Version 0:

Reviewer comments:

Reviewer #1

(Remarks to the Author)

Sissala et al. conducted a study with the aim of comparing the Olink proteomics method (Explore 3072 panel) with mass spectrometry-based TMT-proteomics for plasma proteomics.

The study seems sound and well-performed and is of interest to the field and deserves some credit. Some serious issues should still be addressed. Most importantly:

This section is somewhat redundant as redundant information is already available in the literature. Furthermore, Olink analyses known proteins and the GO results are therefore quite predictable. Furthermore, there are several similar MS-based plasma proteome studies in the literature that have performed GO analyses on found proteins. So this section can easily be removed without losing the quality of the study.

The authors used Spearman correlation analysis to investigate cross platform correlations/precision of the platforms. This method is only suitable for the evaluation of non-normally distributed data. Did the authors check the MS and Olink dataset for normality? If normality, data should be assessed using Pearsons correlation analysis. Another option could be to apply a Bland-Altman which measures the agreement between the two different assays.

The authors used a top14 depletion prior to TMTpro-labeling and MS analysis. Depletion may significantly influence on MS-based protein quantification for several reasons. Please include discussion of the influence depletion on protein quantification in MS-based plasma proteomics in the etc.

A number of standard measurements of e.g. CRP, LDL, HDL must have been performed on the patients included in the study. And some of these are probably also measured with OLink and/or MS. It would strengthen the study if correlations between laboratory data and Olink/MS data are included in the study.

Page 19, line 39-40: The authors highlight the orbitrap astral which shows promising results in plasma proteomics and high throughput. Here, the Tims TOF pro2 should also be mentioned.

The authors applied a Normalized Collision Energy of 30%, which is a bit low for TMTpro leading to lower reporter ion signals, which again leads to less accurate quantification. Moreover, a 2 Da isolation window was applied – a lower isolation windows will reduce co-isolation and reporter ion interference. This should at least be mentioned as issues impacting on quantification accuracy.

Reviewer #2

(Remarks to the Author)

This is a paper that will be of high interest to the community given the recent increase in antibody-based proteomics techniques. The numbers of the comparisons for Olink vs LCMS would ideally be higher and utilization of the currently available 5k Olink panel for a more comprehensive analysis but it is valuable nonetheless. A number of improvements are suggested:

A comprehensive analysis of the current Olink vs LCMS vs Somalogic studies to date including numbers of samples compared as part of the Intro/background part of the paper. This would be valuable in the form of a table such that the value of this study is evident. Comparisons with the recent Coon lab paper (<https://pubmed.ncbi.nlm.nih.gov/39868270/>) would be valuable to determine similarities/differences in findings.

The study compared the correlation between Olink and TMT based LCMS. Since TMT is known to be more variable when there are > 1 plex, it would be valuable to compare the correlations of Olink vs LCMS on just a single plex vs multiple plexes. Precision part (page 9). Assay CVs were evaluated at the protein level. It would also be valuable to assess CV at the peptide level for the technical standards, especially given the PeptOlink tool.

CVs: For the LCMS data it would be valuable to compare the technical CVs for high abundant proteins (that should be depleted) vs those that should not be depleted to assess the added variability that is introduced with the depletion step.

Fig 4: Axes (log₂ fold changes) on the scatter plots should be the same for x and y

Cross-platform correlation of protein levels (page 10): in addition to Spearman ranking, Pearson correlation would be more appropriate to assess the correlation.

MS3 was not used for TMT quant. A discussion of the limitations of the use of MS2 should be added as this is likely going to introduce error.

Olink uses volume of plasma as input for normalization whereas LCMS TMT is using mass based on the author's methods. Were ratio checks performed? It would be valuable to include the variation in TMT channel intensities to be able to assess the variation that is present prior to normalization. It would be valuable for the authors to assess the correlation between TMT LCMS and Olink prior to TMT column adjustment and post.

Reviewer #3

(Remarks to the Author)

This paper is an important contribution to the literature, though it is small in size and more work is needed in future. The work is clearly presented and the Shiny application will make the results more accessible. Analysis of MS/Olink correlation by protein region is particularly impactful (Fig 7 is very interesting/we need more of this kind of analysis to understand what proteoforms affinity reagents are likely measuring). I do have some mostly minor suggestions to improve the work, with one major suggestion about doing more to incorporate missing or below LOD values for proteins into comparison workflow.

Just dropping everything below LOD, as authors do in many of their comparisons, may reduce the number of proteins so much as to not best leverage platform information. It would be great if authors could note concordance of whether a protein is missing or below LOD on both platforms, or just one, not just drop these values and proteins in their analyses. It would be reasonable to also present analyses where missing values from MS are imputed to half minimum or another common approach, and Olink below LOD values retained. This will likely lower overall correlations, but I think is fair and important to present and will be more comparable with much of the prior literature. This is my biggest critique of current methods.

It is potentially problematic that over 20% of proteins don't have CVs calculated- could either the Olink provided NPX values, even if below LOD, be included (this is what is recommended by the company for pQTL analyses etc, values were retained in the UK Biobank proteomics papers).

In the comparisons between previous literature paragraph, are you only limiting to these proteins with essentially no missingness/values below LOD? "On the subset of overlapping proteins with no missing values or QC warnings (N = 463), the median correlation reached = 0.68, with 81% of proteins having a moderate to high correlation between platforms (Fig. 5C, Table S7)." If so, I don't think that's entirely a fair comparison, as those other papers are including a much broader number of proteins. The discussion of these prior studies could be improved to better detail how missing and below LOD values are handled, and how many proteins are compared, in this vs prior studies. Also, the prior studies do have sample sizes- N_Previous.study in the supplemental table shouldn't be NA so often. I think the confusion here may be in large part because below LOD values were retained in those prior Olink/SomaScan comparisons, for the most part. Then sample size is the same for all protein comparisons in a large supplemental table.

Figures S6, 8, 9 (comparisons to prior papers) are I think also included in the supplemental tables, but it is somewhat hard to follow. Could explanations of the column names be included in the supplemental table legends, and the supplemental tables clearly referenced in the figures? Table S17 is quite helpful, but not enough. More expanded table titles and explanations of column names are needed for these parts of the supplement, and some others- if column name not important enough to explain it's likely not required in tables in my opinion.

Important future directions, such as performing such analyses in larger sample sizes so that pQTLs which may influence abundance or may influence antibody binding can be appropriately differentiated for MS platform overlapping proteins, should be mentioned/slightly expanded on in the discussion.

Authors should also cite/explain recent work showing a higher level of below LOD values as the Olink panel expands <https://academic.oup.com/clinchem/advance-article/doi/10.1093/clinchem/hvaf030/8103831>. This has implications for future MS-Olink comparison studies.

Could DAPs identified in this study be assessed for replication using any previously published or publicly available external datasets? Were DAPs from Olink more likely or less likely to replicate, proportionally, than MS in external datasets with similar phenotypes tested (this could at least be examined for males vs females comparison)?

While a preprint, this closely related paper that includes multiple MS sample preparation and analysis platforms should be

more fully discussed or at least mentioned. <https://www.biorxiv.org/content/10.1101/2025.02.14.638375v1.full>

Version 1:

Reviewer comments:

Reviewer #1

(Remarks to the Author)

The authors have responded satisfactorily to the suggestions for improvements. No further comments.

Reviewer #2

(Remarks to the Author)

Recommend to accept. The authors have addressed the majority of the reviewers' concerns.

Reviewer #3

(Remarks to the Author)

In general the revision has cleared up many of my questions. The paper seems close to ready for publication. I am still confused why intra-assay CVs are not available for all Olink assays (n=2913 post QC as in figure 2A) if the below LOD values are now retained, as stated in the response. This needs to be clarified.

Exploration of appropriate MS imputation methods should be mentioned in discussion as a topic for future work. This would allow more comparability in DAP analysis for example.

Minor in Table S10 – should be preprint not ‘preprint’

Since preprints now included- this one should also be in Table S10 and the other corresponding tables- <https://pmc.ncbi.nlm.nih.gov/articles/PMC11844639/>

Reviewers' comments:

We thank the reviewers for their constructive feedback. Please find the responses to specific questions below.

Reviewer #1 (Remarks to the Author):

Sissala et al. conducted a study with the aim of comparing the Olink proteomics method (Explore 3072 panel) with mass spectrometry-based TMT-proteomics for plasma proteomics.

The study seems sound and well-performed and is of interest to the field and deserves some credit. Some serious issues should still be addressed. Most importantly:

This section is somewhat redundant as redundant information is already available in the literature. Furthermore, Olink analyses known proteins and the GO results are therefore quite predictable. Furthermore, there are several similar MS-based plasma proteome studies in the literature that have performed GO analyses on found proteins. So this section can easily be removed without losing the quality of the study.

- We agree that the results from the GO analyses are expected, but we deem them important to keep for three reasons – to demonstrate that the dataset is representative, that the findings reported in other studies are reproducible, and to provide context for readers who might not be familiar with the field. Therefore, we kept them in the manuscript but shortened this section.

The authors used Spearman correlation analysis to investigate cross platform correlations/precision of the platforms. This method is only suitable for the evaluation of non-normally distributed data. Did the authors check the MS and Olink dataset for normality? If normality, data should be assessed using Pearson's correlation analysis. Another option could be to apply a Bland-Altman which measures the agreement between the two different assays.

- We have checked the distribution for both MS and Olink data. Overall, the distribution of values is normal, but many per-protein distributions of values violate the assumption for normality. We have used Spearman correlation coefficient since it does not require normality and is more robust to outliers. In this revised manuscript draft, we have now included both Pearson and Spearman correlation coefficients (Figure S5 and Table S8). Overall, they were similar, but Pearson coefficients were somewhat higher. However, we still based the interpretation of the findings based on Spearman coefficients, to avoid overestimation in instances when the data are not normally distributed, even though it might be slightly more conservative than Pearson coefficients on normally distributed data. Furthermore, Spearman coefficients allows better comparability with published literature, since all but one report Spearman's coefficients.

The authors used a top14 depletion prior to TMTpro-labeling and MS analysis. Depletion may significantly influence on MS-based protein quantification for several reasons. Please include discussion of the influence depletion on protein quantification in MS-based plasma proteomics in the etc.

- Since we report one of the higher MS-Olink correlations, one could speculate that the influence of depletion is rather minor and possibly improves the quantifications by

reducing the peptide complexity. We have also observed a trend of lower CVs for depleted proteins, but the number of depleted proteins was too low to make conclusive remarks -please refer to a similar question raised by reviewer 2 on variations in CVs related to depletion. However, without a proper experimental design that addresses this, we avoided speculating on the effect of depletion further and have now pointed out depletion under technical factors whose impact on the agreement would be interesting to explore in future studies. Please refer to the segment on limitations, under discussion.

A number of standard measurements of e.g. CRP, LDL, HDL must have been performed on the patients included in the study. And some of these are probably also measured with OLink and/or MS. It would strengthen the study if correlations between laboratory data and Olink/MS data are included in the study.

- We agree that this would have been an interesting comparison. Unfortunately, we don't have this clinical data from the participants included in this study. Olink does not include assays that measure CRP but the proteins that constitute the LDL and HDL particles, like APOE, APOB, PCSK9, and APOA1, APOA2, LCAT, respectively, showed varying agreement between the platforms (please refer to Table S8). However, in a previous study we have shown very good agreement between standard clinical immunoassays and HiRIEF relative quantification of CRP, AST, and ALT (Babacic et al. Nature Comms 2023).

Page 19, line 39-40: The authors highlight the orbitrap astral which shows promising results in plasma proteomics and high throughput. Here, the Tims TOF pro2 should also be mentioned.

- This is a good point. We have now changed the discussion and refer to general improvements in MS instrumentation, without necessarily highlighting one instrument. We have further added references for timsTOF along with the Orbitrap Astral MS instrument.

The authors applied a Normalized Collision Energy of 30%, which is a bit low for TMTpro leading to lower reporter ion signals, which again leads to less accurate quantification. Moreover, a 2 Da isolation window was applied – a lower isolation windows will reduce co-isolation and reporter ion interference. This should at least be mentioned as issues impacting on quantification accuracy.

- The high level of prefractionation provided by High Resolution Isoelectric Focusing (HiRIEF) enables sample complexity reduction to great extent, producing 72 compartmentalized fractions of peptide populations (concatenated into 40 fractions). These have little overlap between them (Branca et al. Nature Methods 2014; and Zhu et al. Nature Comms 2018), and the concatenated fractions include smaller peptide populations. Because of this complexity reduction, precursor interference becomes a relatively minor concern. As such, one can employ a wider isolation window than usual, which in turn ensures inclusion of all ions from the isotopic cluster of the precursor ion, resulting in higher signal to noise ratio in the fragmentation spectrum.

Regarding normalized collision energy (NCE), tandem mass tags (TMT)-labelled peptides, and particularly TMTPro labelled peptides indeed require a higher NCE for their HCD than label free peptides do. We have tested these parameters thoroughly with HiRIEF fractionated samples and the here employed parameters are optimized in terms

of identification rates and quantification accuracy. We found that even though higher NCEs did achieve higher reporter ion intensities, they incurred in the loss of larger b and y ions, which in turn led to somewhat poorer identification rates.

Due to manuscript length limitations, we couldn't discuss this in more detail. However, under limitations, we have now included a broader discussion regarding MS factors whose impact on accuracy should be explored in future studies.

Reviewer #2 (Remarks to the Author):

This is a paper that will be of high interest to the community given the recent increase in antibody-based proteomics techniques. The numbers of the comparisons for Olink vs LCMS would ideally be higher and utilization of the currently available 5k Olink panel for a more comprehensive analysis but it is valuable nonetheless. A number of improvements are suggested:

A comprehensive analysis of the current Olink vs LCMS vs Somalogic studies to date including numbers of samples compared as part of the Intro/background part of the paper. This would be valuable in the form of a table such that the value of this study is evident.

- We agree that it would be useful to include this information in the introduction section. Due to the number of studies, this would make the introduction very long and difficult to read. Therefore, in Table S10 we report an overview of the comparable studies that have been used for comparison in this study. We have now further expanded the table and included studies published as preprints, further adding the number of samples per study, the number of proteins analysed in the study, the median correlation for all proteins reported in the study, the number of proteins overlapping with this study, and the median correlation for the overlapping proteins.

Comparisons with the recent Coon lab paper (<https://pubmed.ncbi.nlm.nih.gov/39868270/>) would be valuable to determine similarities/differences in findings.

- We agree with the reviewer and have explored this option, but a direct analytical comparison to our study is not quite possible. Beimers et al. (J Proteome Res 2025) have focused on comparing the number of protein identifications and CVs while we have focused on cross-platform per-protein correlations. We have now included this manuscript in the discussion.

The study compared the correlation between Olink and TMT based LCMS. Since TMT is known to be more variable when there are > 1 plex, it would be valuable to compare the correlations of Olink vs LCMS on just a single plex vs multiple plexes.

- We aimed for a comparison of the entire MS workflow; therefore, we used all TMT sets in which a protein has been identified and quantified. We avoided analyzing single TMT sets, since this would be very much impacted by the sample size and the per-set proportion of missing values (see Figure S1), which would lead to less reliable correlation estimates per TMT set. Since we have shown that the number of missing values is one of the major factors behind worse correlations, we deemed the analyzes per single TMT set less reliable and redundant.

Precision part (page 9). Assay CVs were evaluated at the protein level. It would also be valuable to assess CV at the peptide level for the technical standards, especially given the PeptOlink tool.

- This is a very interesting point. We have added this analysis in Figure 4B. The CVs are overall still relatively low, but somewhat higher than at the protein level. This is expected, since having more peptides for quantifying a protein generally leads to higher accuracy and precision, which is reflected in lower CVs at protein level. However, for the purpose of exploring peptide-Olink correlations, it is very beneficial to see that, overall, HiRIEF LC-MS/MS' precision in quantifying peptides is relatively good for most.

CVs: For the LCMS data it would be valuable to compare the technical CVs for high abundant proteins (that should be depleted) vs those that should not be depleted to assess the added variability that is introduced with the depletion step.

- We have analyzed the depleted and non-depleted proteins separately and observed a trend of lower CVs for the depleted proteins, but found no difference in the correlation estimates between them (please refer to figures below). However, the number of depleted proteins is small, making it more difficult to compare to non-depleted proteins, and depleted proteins are more abundant in plasma, which has likely led to more PSMs and unique peptides that were used for quantification. Both reasons, among others, could have contributed to higher correlation coefficients for the depleted proteins, making the finding of lower CVs among depleted proteins less confident. Therefore, we avoided reporting this in the supplementary figures since it might be better to address this question experimentally in subsequent studies. We have further pointed this out under limitations in the discussion.

Fig 4: Axes (log2 fold changes) on the scatter plots should be the same for x and y

- Thank you for spotting this. We have now fixed it.

Cross-platform correlation of protein levels (page 10): in addition to Spearman ranking, Pearson correlation would be more appropriate to assess the correlation.

- We have now added Pearson correlation coefficients for each protein as well (Figure S5 and Table S8). However, since the assumption of normality was violated in some

instances and Spearman correlation coefficients are more robust to outliers, we have based our interpretation on Spearman correlation estimates. Please refer to a similar question raised by reviewer 1.

MS3 was not used for TMT quant. A discussion of the limitations of the use of MS2 should be added as this is likely going to introduce error.

- Please refer to our response to the question on normalized collision energy, raised by reviewer 1. For the same reasons outlined above, using MS3 on highly pre-fractionated samples might be time-consuming and counterproductive - the risk of isolation interference is relatively minor, while the intensity loss associated with MS3 becomes a more significant drawback. Under limitations, we have now pointed out MS3 as one of the factors that were not evaluated in this study but would be relevant to explore in future studies.

Olink uses volume of plasma as input for normalization whereas LCMS TMT is using mass based on the author's methods. Were ratio checks performed? It would be valuable to include the variation in TMT channel intensities to be able to assess the variation that is present prior to normalization. It would be valuable for the authors to assess the correlation between TMT LCMS and Olink prior to TMT column adjustment and post.

- We agree that evaluating the influence of normalization on the accuracy would be relevant. However, exploring how modifications to our protocol would have affected the quantifications and consequently the agreement is outside of the scope of this study. Therefore, under limitations, we have also pointed out data normalization as future directions.

Reviewer #3 (Remarks to the Author):

This paper is an important contribution to the literature, though it is small in size and more work is needed in future. The work is clearly presented and the Shiny application will make the results more accessible. Analysis of MS/Olink correlation by protein region is particularly impactful (Fig 7 is very interesting/we need more of this kind of analysis to understand what proteoforms affinity reagents are likely measuring). I do have some mostly minor suggestions to improve the work, with one major suggestion about doing more to incorporate missing or below LOD values for proteins into comparison workflow.

Just dropping everything below LOD, as authors do in many of their comparisons, may reduce the number of proteins so much as to not best leverage platform information. It would be great if authors could note concordance of whether a person is missing or below LOD on both platforms, or just one, not just drop these values and proteins in their analyses. It would be reasonable to also present analyses where missing values from MS are imputed to half minimum or another common approach, and Olink below LOD values retained. This will likely lower overall correlations, but I think is fair and important to present and will be more comparable with much of the prior literature. This is my biggest critique of current methods.

- We fully understand that this point was confusing in the manuscript, as we struggled to find the optimal balance between keeping the explanation simple and short versus clear. We fully agree with the reviewer, and we have kept values below LOD and QC warnings for all proteins in all the analyses, unless stated otherwise. However, we have removed

10 proteins that had values <LOD in all samples, but please note that these proteins were not detected in MS data either, so they would not be analyzed anyhow. We have now edited the text and tried to make this clearer. We have also clarified the figure legends to better explain whether and how data were filtered for each analysis.

- In the section on cross-platform correlations, figure 5A shows correlations for all overlapping proteins, with values <LOD and QC warnings retained. These correlations are the ones that we put most emphasis on in the text and mention in the abstract. Therefore, we have presented the least optimistic/lowest correlation estimates as the main result, in accordance with much of prior literature.
- In the section on statistical power, we chose to only include proteins with no missing values or values <LOD because we wanted to compare statistical power and therefore ensure equal sample sizes for MS and Olink data for each protein. We agree that it would also be valuable to present these analyses on proteins with up to 50% missing values, or all proteins that can be analyzed. We have now added this to the manuscript.
- We deemed imputation of MS data not suitable in this case due to reasons outlined below. Therefore, in the differential abundance analysis that we have now added, the sample sizes are no longer comparable between MS and Olink for each protein. So, the identification of DAPs will also be influenced by the detectability of the proteins with MS/Olink, a point that we have added to the results.
If one imputes MS data in the DAA, the results will also be influenced by the imputation method, and we will essentially be evaluating an imputation method rather than the MS method. We agree that evaluating imputation is relevant and something we have been working on in a separate study, but the scope of this manuscript does not allow space for a proper evaluation of this question. In addition, all analyses we have done can tolerate missing values, so imputation can be avoided even if we include proteins with missing values in the analyses.

It is potentially problematic that over 20% of proteins don't have CVs calculated- could either the Olink provided NPX values, even if below LOD, be included (this is what is recommended by the company for pQTL analyses etc, values were retained in the UK Biobank proteomics papers).

- In the original manuscript we removed values <LOD for the CV calculations because it is recommended by Olink. However, we agree with the reviewer and have now replaced the analysis with CVs calculated by including <LOD values.

In the comparisons between previous literature paragraph, are you only limiting to these proteins with essentially no missingness/values below LOD? "On the subset of overlapping proteins with no missing values or QC warnings (N = 463), the median correlation reached $r = 0.68$, with 81% of proteins having a moderate to high correlation between platforms (Fig. 5C, Table S7)." If so, I don't think that's entirely a fair comparison, as those other papers are including a much broader number of proteins. The discussion of these prior studies could be improved to better detail how missing and below LOD values are handled, and how many proteins are compared, in this vs prior studies. Also, the prior studies do have sample sizes- $N_{\text{Previous.study}}$ in the supplemental table shouldn't be NA so often. I think the confusion here may be in large part because below LOD values were retained in those prior Olink/SomaScan comparisons, for the most part. Then sample size is the same for all protein comparisons in a

large supplemental table.

- In the comparisons with previous literature, we included all proteins quantified with both MS and Olink (N = 1129) and retained the values <LOD and QC warnings. We have now clarified this in the text to avoid confusion.
- In Table S10, we have now added the number of samples and proteins overlapping between technologies in each study, as well as the number of proteins overlapping between each individual study and our study. We have filled in N_Previous.study for each study, assuming that it was the total number of samples if not indicated otherwise and if there was no mention of any data filtering in the paper.

Figures S6, 8, 9 (comparisons to prior papers) are I think also included in the supplemental tables, but it is somewhat hard to follow. Could explanations of the column names be included in the supplemental table legends, and the supplemental tables clearly referenced in the figures? Table S17 is quite helpful, but not enough. More expanded table titles and explanations of column names are needed for these parts of the supplement, and some others- if column name not important enough to explain it's likely not required in tables in my opinion.

- We have now moved the supplementary table legends from the supplementary figure file to the Excel file, further explained the column names, and referenced the supplementary tables in the figure legends.

Important future directions, such as performing such analyses in larger sample sizes so that pQTLs which may influence abundance or may influence antibody binding can be appropriately differentiated for MS platform overlapping proteins, should be mentioned/slightly expanded on in the discussion.

- We have now briefly mentioned the importance of cross-validating pQTL findings with both affinity and MS-based methods.

Authors should also cite/explain recent work showing a higher level of below LOD values as the Olink panel expands <https://academic.oup.com/clinchem/advance-article/doi/10.1093/clinchem/hvaf030/8103831>. This has implications for future MS-Olink comparison studies.

- The manuscript has been published during the peer review of our manuscript and we have now included it in the comparison of cross-platform correlations and mentioned in the discussion the findings regarding the higher proportion of values <LOD in Olink Explore HT.

Could DAPs identified in this study be assessed for replication using any previously published or publicly available external datasets? Were DAPs from Olink more likely or less likely to replicate, proportionally, than MS in external datasets with similar phenotypes tested (this could at least be examined for males vs females comparison)?

- We have added a figure on testing the replication rate on differences between sexes, analyzed with different proteomics methods - with MS (one dataset), Olink (two datasets), and SomaScan (one dataset). This revealed that HiRIEF LC-MS/MS had somewhat higher replication rates even though it initially has lower statistical power.

While a preprint, this closely related paper that includes multiple MS sample preparation and analysis platforms should be more fully discussed or at least mentioned. <https://www.biorxiv.org/content/10.1101/2025.02.14.638375v1.full>

- We agree. This is a very valuable study for comparison. We have previously focused on published literature but have now added all the relevant preprints from studies that were ongoing peer review in parallel. We have now also included the preprint above (Kirsher et al., 2025) in the analyses and have briefly discussed their findings.

RESPONSE TO REVIEWERS' COMMENTS:

Reviewer #1 (Remarks to the Author):

The authors have responded satisfactorily to the suggestions for improvements. No further comments.

We thank the reviewer for their constructive and positive feedback.

Reviewer #2 (Remarks to the Author):

Recommend to accept. The authors have addressed the majority of the reviewers' concerns.

We thank the reviewer for their constructive and positive feedback.

Reviewer #3 (Remarks to the Author):

In general the revision has cleared up many of my questions. The paper seems close to ready for publication.

We thank the reviewer for their constructive and positive feedback.

I am still confused why intra-assay CVs are not available for all Olink assays (n=2913 post QC as in figure 2A) if the below LOD values are now retained, as stated in the response. This needs to be clarified.

We have now added a clarification on this under results:

“Data for one control were missing for a subset of assays due to a technical failure, leaving 2,197 protein assays (2,185 unique proteins, 75%) for CV calculation.”

Exploration of appropriate MS imputation methods should be mentioned in discussion as a topic for future work. This would allow more comparability in DAP analysis for example.

We have now added a discussion on imputation methods:

“Lastly, since data completeness affects both the quantitative agreement between platforms and the statistical significance of biological findings, future work would benefit from assessing the comparability of MS and Olink results using imputed data. However, as the outcome of such an analysis would depend heavily on the chosen imputation method, it would require a thorough evaluation of different imputation strategies. Given the high proportion of missing values in global MS data, and the increasing number of values below LOD in Olink data, evaluating imputation methods on different types of plasma proteomics data remains an important future direction.”

Minor in Table S10 – should be preprint not ‘preprint’

Thank you for spotting this typo. We have now corrected it.

Since preprints now included- this one should also be in Table S10 and the other corresponding tables-<https://pmc.ncbi.nlm.nih.gov/articles/PMC11844639/>

We agree and have now included this preprint, along with two other datasets that we identified in a more recent literature search. We have included these in the cross-study analysis of correlations, in the list of previous studies in Table S10 (now Supplementary Data 10) and in Figure S8. The newly added studies are the following:

Nicholas, J. C. *et al.* Cross-Ancestry Comparison of Aptamer and Antibody Proteomics Measures. *Res. Sq.* rs.3.rs-5968391 (2025) doi:10.21203/rs.3.rs-5968391/v1.

Assi, I. Z. *et al.* Correlation between Olink and SomaScan proteomics platforms in adults with a Fontan circulation. *Int. J. Cardiol. Congenit. Heart Dis.* **20**, 100584 (2025).

Wang, B. *et al.* Comparative studies of 2168 plasma proteins measured by two affinity-based platforms in 4000 Chinese adults. *Nat. Commun.* **16**, 1869 (2025).